# Black Garlic and Its Bioactive Compounds on Human Health Diseases: A Review

**DOI:** 10.3390/molecules26165028

**Published:** 2021-08-19

**Authors:** Tanvir Ahmed, Chin-Kun Wang

**Affiliations:** Department of Nutrition, Chung Shan Medical University, 110, Section 1, Jianguo North Road, Taichung 40201, Taiwan; tanvirsust2011@gmail.com

**Keywords:** black garlic, health diseases, health benefits, bioavailability

## Abstract

Black garlic (BG) is a form of aged garlic obtained from raw garlic (*Allium sativum*) via Millard reaction under high temperature (60–90 °C) and humidity (70–90%) for a period of time. Several studies reported higher contents of water-soluble antioxidants compounds (S-allyl cysteine, S-allyl-mercapto cysteine), 5-hydroxymethylfurfural, organosulfur compounds, polyphenol, volatile compounds, and products of other Millard reactions compared to fresh garlic after the thermal processing. Recent studies have demonstrated that BG and its bioactive compounds possess a wide range of biological activities and pharmacological properties that preserve and show better efficacy in preventing different types of diseases. Most of these benefits can be attributed to its anti-oxidation, anti-inflammation, anti-obesity, hepatoprotection, hypolipidemia, anti-cancer, anti-allergy, immunomodulation, nephroprotection, cardiovascular protection, and neuroprotection. Substantial studies have been conducted on BG and its components against different common human diseases in the last few decades. Still, a lot of research is ongoing to find out the therapeutic effects of BG. Thus, in this review, we summarized the pre-clinical and clinical studies of BG and its bioactive compounds on human health along with diverse bioactivity, a related mode of action, and also future challenges.

## 1. Introduction

With the prevalence of chronic diseases and their associated pathological complications, health has become the top of scientific research priorities, with the goal of finding novel foods and tactics to tackle such public health burdens. The scientific literature has recently witnessed flourishing pharmaceutical and nutrition research to use food plants for their diverse health advantages and possible therapeutic uses. A fundamental objective of this nutritional research is to give scientific knowledge to optimize health, and prevent or delay the development of illness. An increasing amount of scientific data suggests that dietary interventions, particularly those that provide high levels of plant foods, can prevent or delay the growth of chronic age-related pathologies by safeguarding the body’s critical physiological systems. The prohibitive effect of plant foods has been investigated as a novel therapeutic remedy for the presence of pharmacologically active substances. Besides, plant foods, and many of their derivative products, i.e., extract, contain various micronutrients (minerals and vitamins), fibers, and bioactive compounds, known as phytochemicals. Notably, plant-based dietary pattern foods have improved different intermediary outcomes with a lower incidence of chronic diseases, including cancer, cardiovascular, neurodegenerative diseases, modulate intestinal microbiota composition and functionality with a positive human health impact [1,2].

In the last few decades, single subgroups of botanical agents such as vegetables, fruits, herbs, and nuts, have been taken into consideration to use as nutrient supplements and recognized for their numerous health benefits, including the prevention of diseases [3,4,5]. Among these, garlic (*Allium sativum*) has been one of the most important dietary sources not only for health benefits but also as a traditional medicinal all over the world [6,7,8,9,10]. Despite all of the health benefits of garlic, the current global consumption of garlic is declining. Some people are unwilling to eat fresh garlic due to its pungent, offending taste and smell, and it can cause gastrointestinal discomfort in some people [11,12]. Therefore, researchers are trying to use raw garlic in different forms such as smocked garlic, garlic oil macerate, garlic juice, supplement pill, garlic powder, etc., to alleviate these discomforts [13,14].

Among the commercially available various garlic processing products, black garlic (BG) has been well known as the fastest-growing health food and is the most studied [15]. Although BG’s origin remains dubious, the product is believed to have been consumed in Korea, Japan, and Thailand from ancient times; in 2014, BG reported an estimated market value of around $94 million [16]. BG is a form of aged garlic obtained from raw garlic (*Allium sativum*) via the Millard reaction under high temperature (60–90 °C) and humidity (70–90%) for a period of time [17,18]. The offensive and harsh odor of fresh garlic are converted into a chewy texture and sweet taste during the aging process [19]. The aging process not only modifies the nutrients and sensory attributes of BG but also improves bioactivity. The health benefits of BG intake are largely associated with the presence of bioactive substances such as phenol and organosulfur. The bioactive components of BG display a wide variety of physiological functions in the human body, which may have therapeutic potential for treating various diseases. For instance, multiple studies showed the bioactivities of BG, including antioxidation [20], anti-inflammation [21,22,23], anti-obesity [24,25], hepatoprotection [26,27,28], hypolipidemia [29,30], anti-cancer [31,32,33], anti-allergy [34,35], immunomodulation [15,36], cardiovascular prevention [37,38], and neurodegenerative protection [39].

In the last few decades, substantial studies have been conducted on BG and its components against different common human diseases. Still, a lot of research is ongoing to find out the therapeutic effects of BG. However, a review of the health benefits of BG and its components on different common human diseases from fundamental to clinical studies is still lacking. Thus, in this review, we summarize the pre-clinical and clinical studies of BG and its bioactive compounds on human health diseases along with diverse bioactivity, the related mode of action, and also future challenges. We hope a comprehensive overview of published findings on the health benefits of BG and its bioactive compounds can help food and nutrition experts, researchers, and scientists define future research direction with clinical evidence to improve people’s health and wellness.

## 2. Data Collection

The authors of this review article carried out a literature search for relevant articles regarding BG effects on human diseases by determining sources or literature in the form of primary data or official books, national or international journals published till May 2021. Additionally, data searches were also conducted using different online platforms. During writing this review article, the main references were cited from the trusted source, such as *Google Scholar, Web of Science, Scopus, NCBI, Science Direct, ResearchGate, Medline (PubMed*), and other trusted journals publishers. 

## 3. Thermal Processing of Black Garlic

The thermal processing of BG from raw garlic has been processed under controlled temperature and relative humidity without additives, which is ready to consume with a sweet taste and slightly pungent odor and taste. Currently, there is no standard procedure for producing BG; processing conditions vary greatly depending on regional traditions and the specific features desired in the ultimate product. Till now, researchers are trying to find out optimal conditions for BG processing and they have already applied different thermal processing plans (Table 1). BG is commonly processed in temperature ranges between 30–90 °C, 50–90% relative humidity, and an incubation period of 10–80 days [16,40,41,42,43,44,45,46,47]. 

Treatments performed prior to the thermal processing of raw garlic to produce BG are referred to as pretreatments and have been shown to impact the attributes of final BG products. Few garlic pretreatments can damage or destroy the garlic’s cell structure, allowing immediate reactions between the cell components and external elements during the processing of fresh garlic to BG. The temperature and relative humidity of thermal treatment during BG’s production are the critical variables among the parameters that influence the quality of BG. For example, the change in the color of garlic was sluggish and not completely black at 80% relative humidity, and the three days oven drying time at 60 °C; the color change at 70 °C is substantially more than at 60 °C; black color becomes homogenous when heated to 70–80 °C; at 90 °C, the strong flavor of bitterness and sour taste would be noticeable. BG has a dry look at 70 °C with improved flexibility and quality; bone dry at 80 °C, combustion; very harsh at 90 °C, with a distinct scent of burning. BG has optimum softness and elasticity in the case of the moisture level of 400–500 g/kg. On the other hand, BG would become considerably drier with poor elasticity when the moisture content reaches in the range of 350–400 g/kg. In addition, BG would be highly droughty and hard under 350 g/kg of moisture content [41]. 

## 4. Composition of Black Garlic 

Carbohydrates (36%), fat (0.5%), and protein (6%) are the major components of fresh garlic. The carbohydrates comprise fructan (23.2–27. 8 g/100 g), sucrose (0.6–0.7 g/100 g), fructose (0.1 g/100 g), glucose (0.04–0.05 g/100 g). Another study also showed that garlic contained 60% water, 28% carbohydrate, 8.4% protein, and 0.1% fat [11,48,49]. Compared to garlic and BG, when the garlic was prepared at 60 °C and 90% relative humidity for 45 days, the weight of the garlic reduced by 64%, the soluble solid content (°Brix) increased by 13%, and the water activity declined slightly from 0.97 to 0.93. Besides, moisture content was also reduced one to two times compared with garlic and BG. In addition, the carbohydrate profile of fresh garlic has been drastically changed upon its conversion to BG. For instance, carbohydrate content in the BG compared to fresh garlic was increased by 1- to 2-fold followed by 1.3–1.6 fold, 6–108 fold, and 2–13 fold in the case of sucrose, fructose, and glucose, respectively [11]. When the garlic was prepared for 45 days at 60 °C and 90% relative humidity the carbohydrate content increased by 28–47%. During the conversion of fresh garlic to BG by thermal processing, fructans disintegrate progressively into monosaccharides (mostly glucose and fructose), disaccharides, and oligosaccharides by thermally induced degradation and enzymatic cleavage catalyzed by fructan exohydrolase [50,51]. Yuan and his colleagues also obtained similar results in their study and noticed that the fructan levels of garlic were decreased by around 6-fold at 55 °C incubation with 80% humidity for 90 days [52]. In BG, monosaccharides became the predominant saccharides, with minor levels of polysaccharides and disaccharides [53]. BG was detected in most of the saccharides and related products at 60–80 °C for 36 days, whereas a few were found in fresh garlic [54]. According to several studies, polysaccharide breakdown is mostly caused by heat treatment rather than enzymatic hydrolysis, with polysaccharide molecular weight distribution and breakdown rate closely related to processing temperature [55]. However, BG could have a reducing sugar content 20–70 times higher than its original fresh garlic depending on the process of fresh garlic, which causes a distinct sweet taste in BG [41,56].

The lipid contents of fresh garlic and BG also contribute considerably to their sensory properties and serve as a nutrient and energy source. Lipids derived from fresh garlic with chloroform-methanol reportedly constitutes 63% neutral fluids, 14% glycolipids, and 24% phospholipids [57]. Changes in lipid profiles are identified in the development of BG due to their oxidation and engagement in the different chemical processes. Choi et al. [40] demonstrated a notable rise in crude lipid content (0.15% to 0.60%) following fresh garlic processing to BG. In comparison, Lu [58] observed a considerable rise in crude lipid levels of fresh garlic and BG, respectively, of 0.33% and 0.16%. These inconsistencies are most likely caused by the garlic variety, processing technique, extraction, and analysis processes as well as moisture content variations of fresh garlic and BG. In the transformation of fresh garlic to BG, hydrolytic and oxidative alterations are conceivable for garlic lipids under high temperature (50–90 °C) and high humidity (60–90%). A number of chemicals are therefore formed including alcohols, lactones, aldehydes, and ketones. These compounds, together with fatty, take part in a range of complex chemical processes involving Millard reaction, oxidation, and hydrolysis [59].

Fresh garlic is abundant in free amino acids and asparagine, glutamic acid, valine, lysine, and tryptophan are the most predominant free amino acids available in fresh garlic [60]. The processing of BG impacts secondary, tertiary, and/or quaternary protein structures, but is unlikely to disintegrate new free amino acids from covalent peptide bonds. When fresh garlic is converted to BG at high temperatures, protein is denatured, and some free amino acids participate in the Millard reaction. The amino acid profile varies significantly depending on the conditions under which the fresh garlic is converted to BG. For example, Lu [58] found a decrease in the overall amount of 18 free amino acids from 2000 to 1400 mg/100 g fresh matter (FM) with increased levels of leucine, isoleucine, valine, phenylalanine, cysteine, alanine, and aspartic acid. Another study showed the contents of branched amino acids such as leucine (60–70 mg/100 g) and isoleucine (58–50 mg/100 g) were increased in BG when compared with fresh garlic at 70 °C and 90% relative humidity [46]. In addition, several amino acids, especially tyrosine and cysteine, decrease in content due to the link with the Millard reaction—possibly being closely connected with changes in the antioxidant activity when fresh garlic is transformed into BG [61].

Garlic is one of the most abundant sources of phenolic compounds among the vegetables consumed by humans. The total quantities of phenolic compounds in fresh garlic range from 3 to 11 mg gallic acid equivalent (GAE)/g dry matter (DM) with a mean level of 6.5 mg GAE/g DM. In addition, the overall concentration of phenolic acid varies from 2 to 20 mg/kg DM with an average of 7.6 mg/kg DM and caffeic acid is the most predominant content (average 3 mg/kg DM) [62]. The polyphenol level was enhanced by 2.8% with the antioxidant activity (6.7%) when fresh garlic was prepared at 60 °C and 90% relative humidity for 45 days. Phenol-like substances were increased 3- to 4-fold in comparison to fresh garlic and BG, whereas flavonoid-like compounds and antioxidant compounds increased by 2- to 8-fold and 3- to 7-fold, respectively. In general, the polyphenol contents of BG are three times greater in whole BG bulbs as compared to fresh garlic bulbs, and six times greater in BG cloves peeled [18,35,43,46,63]. As a result, BG exhibits greater antioxidants than fresh garlic. However, processing fresh garlic for a long duration at high temperatures could lead to a drop in certain phenolic substances due to the loss of polyphenols that are freshly generated or released [18].

Recently, an amino acid-containing molecule S-allyl cysteine (SAC) has received more attention since it is the most predominant antioxidant of BG and is a biologically active and health-beneficial compound found in both fresh garlic and BG [64]. SAC in fresh garlic was reported between 21 to 23 µg/g FM and could be considerably higher, three to six times as much as fresh garlic, depending on the thermal treatment. Bae et al. [16] found in their study that the SAC content of BG was 124.67 µg/g DM when manufactured at 40 °C for 45 days, but the SAC content decreased to 85.46 µg/g DM in the BG when the temperature raised to 85 °C. Another study reported the higher SAC content in the BG and also noticed that when the fresh garlic was heated at 60 °C and 90% relative humidity for 45 days, the amino acids and SAC levels are enhanced three and eight times, respectively [49].

Similarly, hydroxymethylfurfural (HMF) is one of the major antioxidant ingredients in BG. Besides, the black color formation of the garlic sample is related to the formation of HMF. Zhang et al. [41] processed garlic to BG at a temperature of 60 °C, 70 °C, 80 °C, and 90 °C and monitored the HMF concentration till maturity. When HMF reached 4 g/kg, samples of garlic were mainly aged and turned black. 90 °C-processed samples attained maturity around 9 days, 80 °C-processed samples in approximately 21 days, and 70 °C-processed samples in about 33 days, and at 60 °C the HMF concentration increased very slowly throughout the procedure, resulting in a lower-quality garlic product. HMF levels were sharply increased yet the color and flavor of garlic samples decreased with high temperatures. When the BG production process was carried out at a temperature of 70 °C, the highest-grade items were achieved. In another study, Liang et al. [51] demonstrated that during the processing of fresh garlic in BG, a substantial amount of HMF was generated and the BG extracts acquired in 90 days of thermal processing had an increased HMF content by more than six times, compared to those heated in 25 days. In addition, Li et al. [65] observed that the HMF content in BG could be increased up to 25% by freezing treatment.

Citric acid, which is the most abundant in organic acid, is also found in garlic in relatively large quantities compared to other organic acids such as lactic acid, fumaric acid, malic acid, and formic acid [51,66]. In a study, Bae et al. [16] reported a pH decreased from 6 to 5 or 3 after heating the fresh garlic at 40 °C or 85 °C for 45 days. Zhang et al. [41] observed in their study that the total acid content had been modified from 4.5 g/kg to 33.50, 37.50, 30.90, and 36.35g/kg, respectively, during the transformation of fresh garlic to BG at 60 °C, 70 °C, 80 °C, and 90 °C. Moreover, Liang et al. [51] found that after the fermentation process, the BG extracts newly formed a higher amount of formic acid and acetic acid. Such changes in organic acid are important because a higher organic acid content not only provides a sweet-sour flavor, it also accelerates the breakdown of proteins, polysaccharides, and the microbiological stability of BG samples [67,68]. The increased acidity of fresh garlic after the heat treatment is mainly due to the degradation of a large number of alkaline groups and the formation of short-chain carboxylic acids during reactions such as the Millard reaction [69].

Fresh garlic also contains a number of enzymes, which affect the quality of the final BG products. Superoxide dismutases (SODs) are possibly a class of enzymes found exclusively in garlic. These enzymes are not only important free radical scavengers in a growing microorganism, but they also perform crucial biological functions. Sato et al. [63] revealed that fresh garlic had SOD-like activities; 80% ethanol extract from garlic fermented for 40 days at 60 to 70 °C and 85% to 95% relative humidity exhibited a 13-fold increase in SOD-like activities. Furthermore, fresh garlic also contains multiple water-soluble vitamins and trace elements. The fresh garlic processing at different temperatures and humidity would increase 1.15 to 1.92 times more in total water-soluble vitamins. The garlic samples treated for 60 h at 70 °C and 60% relative humidity showed the highest amounts of water-soluble vitamins (11,600 mg/kg) than fresh garlic (6633 mg/kg) [70]. In addition, garlic also provides a remarkable amount of several minerals ranging from 0.70% to 0.80% of FM [71,72]. Kang reported that different thermal treatments significantly altered the mineral profile of garlic and found that manufacturing garlic for 192 h at 65 °C and 50% relative humidity increased the selenium content with sodium, iron, and calcium [73].

## 5. Formation of Phytochemicals of Black Garlic during Millard Reaction

BG has become one of the world’s fastest-growing health products with increased awareness about the health benefits of garlic. Thermal techniques are frequently employed in food production. One of the major goals of thermal procedures is to increase the sensory quality, palatability, and diversity of colors, aroma, taste, and textures of food. Furthermore, the thermal process produced biological compounds, which are not initially present in foodstuffs (Figure 1) [74]. When BG goes through a natural fermentation or aging process, the garlic cloves turn from white to brown and then finally black due to the Millard reaction. The Maillard reaction alternatively refers to as the amino-carbonyl reaction, is a non-enzymatic process that occurs between reducing sugars and the amino group found in amino acids, peptides, and proteins [75]. In addition, the Maillard reaction was classified into three phases, the first of which began with the reaction between the carbonyl group and reducing sugars and the amino groups of amino acids and continued until the Amadori products (i.e., glucose) or Heyn’s products (i.e., fructose) were formed. The intermediate stage involves the fragmentation of sugars and the degradation of amino acids, which resulted in the formation of a variety of intermediates, most notably di-carbonyl compounds. At the final stage, the intermediates are polymerized, resulting in the development of high molecular weight products, particularly melanoidins (Figure 1) [76].

The concentration of Amadori and Heyns compounds in garlic was determined by ion-pair RP-HPLC-UV in BG and dehydrated garlic as a product of acid hydrolysis such as N-ε-2-furoylmethyl-amino acids (2-FMAAs). The total levels of the three major Amadori compounds such as fructose-proline, fructose-leucine, fructose-valine, and Heyns compounds, i.e., glycosyl-proline, glycosyl-leucine, and glycosyl-valine, varied between 763.1 and 281.5 g/g of product in five samples of Chinese commercial BG. In the case of 2-FMAAs, 2-Furoylmethyl-Lysine, 2-Furoylmethyl-Arginine, and 2-Furoylmethyl-γ-Aminobutyric acid were identified in commercial BG in the range between 62.5 and 144.90 mg 2-FMAAs/100 g protein with furosine being the most prevalent [77]. These furoylmethyl amino acids, particularly furosine, are generated after acid hydrolysis of Amadori and Heyns products and in the early stage of Millard reaction, these compounds are recognized as indicator compounds.

On the other hand, during the manufacturing process of BG, 5-hydroxymethyl furfural (5-HMF) other derivatives such as 5-(hydroxymethyl-2)-furoic acid (5-HMFA) are also developed during the intermediate stage of the Millard reaction as a result of Amadori and Heyns product degradation. During the aging process of BG, the content of 5-HMF, and 5-HMFA may increase [51]. However, the 5-HMF formation is associated with the black color intensity. The color of the garlic turns black when the 5-HMF content reaches around 4 g/kg. In this context, Zhang et al. [41] showed similarity as they obtained the maximum concentration of 5-HMF around 5 g/kg after the manufacturing process of BG. Similarly, Sun and Wang [42] reported a higher content of 5-HMF up to 7 g/kg at high temperature (75 °C) and relative humidity (85%) for 12 days (Table 2).

Melanoidins are carbohydrate- and nitrogen-based polymers with a brown color, high molecular weight, and varying degrees of polymerization that are frequently generated in the late stages of the Maillard reaction during the preparation of BG. The overall contents of melanoidin increased by thermal processing of fresh garlic into BG from trace level to a high level with a concurrent increase in brownness (Table 2) [52]. In addition, few important intermediates such as furan, thiophene, and pyrazine, which give a more intense and desirable aroma in BG, are also produced with melanoidins [78].

During the production of BG by heating at 60–90 °C for 10–15 days, allicin was reduced by 70–80% [92]. This is because of the synthesis of allicin and the thermolysis process during the first two days. Alliin is particularly prone to deterioration due to its unstable sulfoxide bond at a high temperature. Zhang et al. [41] showed the level of alliin declined more quickly at 90 °C than 80, 70, and 60 °C, respectively, after about 12, 24, 33, and 66 days of incubation with nearly the same amount of allicin (0.3 g/kg). Furthermore, Chen et al. [93] explored thermal alliin degradation over 50 h at 60–90 °C and identified secondary organosulfur compounds in the HPLC-MS system, mainly including SAC, di-alanine tetra-sulfide, di-alanine disulfide, allyl alanine disulfide, allyl alanine tetra-sulfide, allyl-alanine trisulfide, and di-alanine trisulfide. SAC is the major organosulfur molecule, which is generated by γ- glutamyl-S-allyl cysteine (catalyzed by γ-GTP) during the enzyme hydrolysis [94]. Although γ-GTP activity might have a chance to be damaged by heating, Chen et al. [93] showed a technique to produce SAC from alliin by direct thermal processing even if γ-GTP was inactive. Recently, Al-Sheri et al. [86] reported 42.7 µg/g DM of SAC in fresh garlic extract and this value increased up to 656.5 µg/g DM in BG extract. According to Sasaki et al. [90], SAC contents increased over 8-fold in BG when compared with fresh garlic. Similarly, Park et al. [87] showed the SAC contents of about 2.5 mg/g dry weight (DW) in fresh garlic and the content increased to 8.05 mg/g DW after the aging process. In another study, Kim et al. [22] quantified the SAC contents of 73.5 ± 12.5 µg/g DW in fresh garlic juice, while in BG juice it was 242.3 ± 6.1 µg/g DW. In addition, Thao et al. [88] and Bae et al. [16] determined higher contents of SAC at a concentration of 124.67 ± 1.61 µg/g and 427.05 ± 3.56 µg/g, respectively. On the other hand, allicin is a highly reactive compound with limited thermal stability. It decomposes into diallyl sulfide (DAS), diallyl disulfide, diallyl trisulfide, ajoene, and finally is transformed into S-allylmercapto-cysteine (SAMC). SAMC is an odor-free, tasteless, and stable molecule that is around 6-fold higher in BG than in fresh garlic [92]. SAMC production in BG is also associated with the γ-GTP enzyme which requires an optimum temperature of 40 °C for activity [16]. Therefore, complete inhibition of the γ-GTP enzyme during BG production is expected. Thus, these findings indicate that the rise in SAC concentration is primarily due to allicin conversion.

Regarding variations in total phenolic content, most of the studies agreed that total phenolic content would increase substantially in BG manufacturing so that the levels of BG polyphenols are multiple times higher than the level of fresh garlic. In a study, Kim et al. [82] demonstrated that the total phenolic contents of BG were found to be approximately 3- to 10-fold higher than fresh garlic and hydroxycinnamic acid derivatives were identified as the major phenolic acids in BG at various processing stages. Similarly, Toledano-Medina et al. [44] showed higher contents of total phenolic content in BG (14,900 mg GAE/kg DM) which was approximately three times higher than fresh garlic. A similar higher tendency of phenolic contents in BG was also noted in the study of Li et al. [65], who reported 24,050 mg GAE/kg of total phenolic content in BG. In addition, similar to polyphenols, the total flavonoid contents in BG may also fluctuate due to processing conditions. The total flavonoid content in BG was increased by 3- to 12-fold compared with that in fresh garlic and flavonols as well as flavonols were identified as major flavonoid compounds in BG [22,43,46,83,84,85].

Furthermore, Yang et al. [75] identified nine key aroma-active compounds in BG including acetic acid (sour), furaneol (caramel), allyl methyl trisulfide (cooked garlic), (*E*,*Z*)-2,6-nonadien1-ol (cucumber), diallyl disulfide (garlic), diallyl trisulfide (sulfur), 5-heptyldihydro-2(3H)-furanone (apricot), 3-methylbutanoic acid (sweat), and diallyl sulfide (garlic) through aroma recombination and omission experiments. Moreover, several studies also reported a higher content of carbohydrates such as fructose, glucose, followed by tryptophan, valine, glycine, and arginine as the major amino acids, saponin, thiosulfate, and pyruvate [18,20,73,76,92].

## 6. Impact of Black Garlic on Health Promotion and Diseases Treatment

BG and its derivatives have been documented to have multiple biological impacts on health promotion and treatment of various diseases. The transformation of fresh garlic to BG results in modifications in the biological activity of bioactive compounds caused by fermentation. In the past few decades, extensive pre-clinical investigations have shown the therapeutical potential of BG against a wide variety of human diseases. Furthermore, it has been demonstrated that BG interacts directly with a number of signaling molecules. These pre-clinical investigations have established a good foundation for clinical trials investigating BG’s efficacy. The pre-clinical and clinical findings on the bioactive effects of curcumin are described briefly in the following sections.

### 6.1. Effects of Black Garlic on Metabolic Disorders

#### 6.1.1. Black Garlic and Diabetes Mellitus

Diabetes mellitus is a heterogeneous form of metabolic disorder characterized by chronic hyperglycemia with impaired carbohydrate, lipid, and protein metabolism resulting from defects in insulin secretion, insulin action, or both [95]. Clinical and experimental studies have demonstrated that chronic hyperglycemia is a major source of oxidative stress and that elevated free radicals play a key role in diabetes mellitus pathogenesis and complications. Several studies have investigated the anti-diabetic potential of BG and these studies described variable effects of BG on conventional diabetes mellitus markers (Table 3). For example, aged black garlic extract exhibited ameliorative action on glyco-metabolic biomarkers in streptozotocin-induced diabetic rats by significantly decreasing blood glucose, glycated hemoglobin, and markedly increased serum insulin. Additionally, aged garlic extract significantly attenuated the elevation of serum triglyceride, total cholesterol, and lowered lipid peroxidation in liver and kidney tissues [96]. A similar type of result was also reported in the study performed by Thomson et al. [97] and Seo et al. [30], where aged garlic extract reduced the oxidative stress markers, improved insulin sensitivity, dyslipidemia, and other complications of diabetes in male Sprague-Dawley rats and C57BL/KsJ-db/db mice, respectively. Another study examined the anti-diabetic effects of BG powder in male Wister rats and found that BG powder lowered blood glucose, prevented glycogen in the liver, and improved lipid metabolism by increasing the activity of glutamic oxaloacetic transaminase, glutamic pyruvic transaminase, and γ-GTP [98]. Moreover, Lee et al. [99] observed that aged garlic extract significantly decreased thiobarbituric acid reactive substances levels and elevated the activities of SOD, glutathione peroxidase (GSH-Px), and catalase (CAT) in diabetic mice. The results obtained by Si et al. [100] demonstrated that 40 weeks of supplementation of BG lowered the levels of plasma malondialdehyde, SOD, GSH-Px, and total antioxidant capacity in pregnant women. They also mentioned that *Lactobacillus bulgaricus* improved the antioxidant capacity of BG in the prevention of gestational diabetes mellitus. On the other hand, Kim et al. [89] used SAC-enriched BG juice and treated streptozotocin 239-induced insulin-deficient mice. The authors found that BG juice improved the glutathione antioxidant system, increased leptin, and adiponectin secretion, inhibited hepatic gluconeogenesis, and suppressed nuclear factor-kappa β (NF-κβ)-mediated inflammatory signaling.

#### 6.1.2. Black Garlic and Obesity

Obesity has become a major public health problem. Obesity-related problems are linked to a variety of metabolic syndrome symptoms including high blood pressure, dyslipidemia, insulin resistance, and glucose intolerance. Various approaches might be employed in human obesity prevention and control. Since a perfect cure or prevention of obesity still remains to be established and the majority of the anti-obesity drugs could have adverse effects, an exploration into the identification of new materials has shown a growing interest. BG extracts have been reported for their activity in reducing body weight, adipose tissue mass, serum triglyceride, total cholesterol, low-density lipoprotein, and plasma malondialdehyde in mice with high-fat-diet-induced obesity (Table 3) [25,29,101]. The fermentation of garlic by lactic acid bacteria ameliorated diet-induced obesity in db/db C57BL/6J mice by decreasing body weight (18%), lowered epididymal (36%), retroperitoneal (44%), and mesenteric adipose tissue mass (63%), respectively. Moreover, fermented garlic extract by lactic acid bacteria also downregulated mRNA protein expression of proliferator-activated receptor γ (PPARγ), CCAAT-enhancer-binding proteins, and lipogenic proteins, including sterol-regulatory element-binding protein-1C (SREBP-1C), fatty acid synthase (FAS), and stearoyl-CoA desaturase-1 (SCD-1) [102]. Similarly, in another study, Seo et al. [103] proposed a mechanism underlying the anti-obesity function of aged garlic, which was closely related to upregulate adiponectin, peroxisome proliferator-activated receptor α, and downregulate SREBP-1C, acetyl-CoA-carboxylase, FAS, and SCD1 in high-fat-diet mice. Xu et al. [104] carried out a clinical trial on 51 healthy adults for 6 weeks to investigate the effect of aged garlic extract on chronic inflammation and immune function in adults with obesity. A daily dose of 3.6 g aged garlic extract prevented the increase of serum tumor necrosis factor α (TNF-α), interleukin-6 (IL-6), and reduced the levels of blood low-density lipoprotein in adult obesity. Moreover, Chen et al. [105] revealed that methanolic extract of BG ameliorated diet-induced obesity in male Wister rats via upregulate AMP-activated protein kinase, Forkhead box O1, sirtuin 1, adipose triglyceride lipase, hormone-sensitive lipase (HSL), perilipin, acyl-coenzyme A oxidase, carnitine palmitoyl transferase 1, uncoupling protein 1, and downregulated CD36 in the adipose tissue. Aged black garlic extract has also been reported to show anti-lipogenic and lipolytic effects in mature 3T3-L_1_ adipocytes by reducing protein expression of PPARγ, HSL, and serum-phosphorylated HSL levels [106].

### 6.2. Effects of Black Garlic on Genitourinary Tract Diseases

BG has also demonstrated an intriguing capacity to alleviate symptoms associated with a variety of genitourinary diseases. Only a few research have examined the role of aged garlic extract in the treatment of kidney disorders, more precisely those caused by glomerular, tubulo-interstitial, or infectious pathologies (Table 4). In 2003, Maldonado et al. [108] demonstrated the effects of aged garlic extracts on gentamicin-induced nephrotoxicity in male Wister rats. It was found that aged garlic extract prevented gentamicin-induced nephrotoxicity by decreasing the oxidative stress and preserving the activities of manganese superoxide dismutase, glutathione peroxidase (GPx), and glutathione reductase (GR). In the case of diabetic nephropathy disease, aged garlic extract significantly decreased albumin levels in urine, blood urea nitrogen contents, and increased urine urea nitrogen contents in diabetic rats. The protective effect of aged garlic extract on diabetic nephropathy may be due to its anti-glycation and hypolipidemic effects [109]. Albrakati et al. [110] studied the effect of the aged black extract on chronic kidney disease and found that aged garlic extract rescued ethephon-induced kidney damage through the activation of nuclear factor erythroid 2–related factor 2 and inhibition of inflammation and apoptotic response. Another study performed by Lee et al. [111], on acute kidney injury in rats, showed very promising results when treated with aged black garlic extract. Treatment with aged black garlic extract prevented deterioration of renal function monitored by standardized biomarkers such as the serum levels of creatinine and blood urea nitrogen. Levels of oxidative stress markers such as 8-hydroxydeoxyguanosine, malondialdehyde, NF-κβ, inducible nitric oxide (NO) synthase, cyclooxygenase-2 (COX-2), and transforming growth factor-beta 1 (TGF-β1) were lower in the aged black garlic.

### 6.3. Effects of Black Garlic on Digestive Diseases

#### 6.3.1. Black Garlic and Liver Diseases

Numerous substances, including dietary components, drugs, alcohols, and pollutants, can cause acute and/or chronic liver diseases, such as liver fibrosis, non-alcoholic liver disease, non-alcoholic steatohepatitis, and even cirrhosis. BG has been widely investigated for its hepatoprotective properties (Table 5). In Wister rats with CCl_4_-induced liver injury, the administration of BG extract was found to show a hepatoprotective effect based on the protection against oxidative damage. The levels of SOD, GSH-Px were decreased with an increase in the levels of alanine transaminase (ALT), aspartate transaminase (AST), lactate dehydrogenase, and alkaline phosphatase (ALP) [112]. In another study, Tsai et al. [28] demonstrated that the administration of SAC and polysaccharides enriched-BG extract into ICR mice inhibited CCl_4_-induced hepatic injury by inhibiting lipid peroxidation and inflammation. Moreover, Kim et al. [113] demonstrated the hepatoprotective effects of aged black garlic on chronic-induced liver injury in Sprague-Dawley rats due to the suppressive effects of aged black garlic extract on cytochrome P450 2E1 activity and its inductive effect on glutathione-s-transferase and quinone reductase activities. Shin et al. [27] reported that the administration of aged black garlic extract lowered the AST and ALT levels in the liver of Sprague-Dawley rats treated with CCl_4_ and D-galactosamine, consequently protecting the liver. The hepatoprotective effects of fermented garlic extract were also confirmed in the two different studies carried out by Jiang et al. [114] and Chung et al. [26], where the administration of fermented BG extract into C57BL/6 mice modulated glycometabolism, lipometabolism, oxidative stress, and inflammation. Therefore, BG may be a promising agent to prevent oxidative stress and cholesterol-related liver disorders, by decreasing the levels of ALT, AST, ALP, total cholesterol, low-density lipoprotein-cholesterol (LDL-C), and malondialdehyde, increasing SOD, GSH-Px, CAT, GPx, and GR. Furthermore, Lee et al. [115] showed that lactic acid-fermented garlic extracts protected against oxidative liver injury by acetaminophen through inhibiting apoptosis, maintaining cellular GSH, and protecting from oxidative damage to mitochondria as well as suppressing liver mitogen-activated protein kinases (MAPKs) activation.

#### 6.3.2. Black Garlic and Inflammatory Diseases

Numerous in vitro investigations have demonstrated that BG has significant potential for treating a variety of diseases related to inflammation such as lethal sepsis, endometriosis, rheumatoid arthritis, inflammatory bowel diseases, etc. (Table 5). In lipopolysaccharide (LPS)-induced RAW 264.7 cells, the isolated 2-linoleoylglycerol from aged garlic was found to be capable of reducing the levels of NO, inflammatory mediators such as IL-6, TNF-α, and interleukin-1β, prostaglandin E2 as well as the expression of COX-2 and inducible nitric oxide synthase (iNOS). In this context, the anti-inflammatory action of aged garlic extract is most likely to suppress the expression of classical MAPKs (extracellular-signal-regulated kinase (ERK) and p38 MAPK) [22]. A similar type of results was also reported by Park et al. [116], who observed that aged garlic extract inhibited the expression of COX-2 and prostaglandin E2 production by phorbol 12-myristate-13 acetate through inactivation of NF-κβ in the human pre-monocytic cell model (U937). Kim et al. [117] reported that 5-HNF isolated from aged black garlic prevented TNF-α-induced monocytic cell adhesion to human umbilical vein endothelial cells (HUVECs) via suppression of vascular cell adhesion molecule-1 (VCAM-1) expression, reactive oxygen species generation (ROS), and NF-κβ activation. In another study, 5-HNF exerted anti-inflammatory activity in RAW 264.7 cells against LPS-induced inflammatory response through inactivation of MAPKs, NF-κβ, and intracellular signaling Akt/m TOR pathway [118]. In vitro, a hexane extract of aged black garlic inhibited cell proliferation and cell cycle progression in TNF-α activated human endometrial stromal cells through the downregulation of ICAM-1 and VCAM-1 expression and IL-6 secretion by inhibiting the activation of NF-κβ, ROS formation, activator protein-1 and the ERK as well as c-Jun N-terminal kinase signaling pathways [119]. In RAW 264.7 macrophages, it was shown that the anti-inflammatory effects of aged black garlic may be due to the inhibition of MAPKs, NF-κβ activities, cytokine production, and expression of iNOS and COX-2 [21,120].

#### 6.3.3. Black Garlic and Other Gastrointestinal Diseases

BG is a long-term dietary and medicinal supply capable of regulating the gastrointestinal system and promoting digestion. BG polysaccharide has been reported to show great potential in promoting gastrointestinal health benefits (Table 5). Chen et al. [121] recently evaluated the effect of BG extract on gastrointestinal motility. Results showed that in vitro, the BG n-butanol fraction extract exerted a substantially more significant impact in the small intestine. In addition, an increase in the levels of 5-hydroxytryptamine receptor 4 was efficient in stimulating gastrointestinal peristalsis, enhancing the emptiness of its gastrointestinal tract and defect promotion. Very few have studies investigated the laxative effects of BG. Li et al. [122] prepared a mixed BG beverage and the administration of high, medium, and low-dose BG beverage into BALB/c mice demonstrated that mixed BG beverage improved the intestinal flora of mice. The authors of this study concluded that a medium dose of mixed black garlic beverage showed a relatively high ink-propelling rate with less defection time in comparison with the negative group. Besides, the amounts of *E. coli* and *Enterococci* slightly increased but *Lactobacillus* had a significant difference before and after stomach perfusion. Recent studies reported that BG can be used therapeutically for inflammation-based gastroesophageal reflux disease including reflux esophagitis. Kim et al. [123] conducted a study to evaluate the protective effect of SAC-enriched BG on reflux esophagitis (RE) in Sprague-Dawley rats. BG was given an oral dose of 100 mg/kg body weight two hours before RE induction and its effects were compared to those of raw garlic. The authors of this study found that BG, rather than raw garlic, significantly inhibited RE-induced histological changes. The reduced catalase was dramatically increased by BG supplementation; nevertheless, levels of SOD tended to increase in the esophagus. These results suggested that BG treatment can ameliorate the development of esophagitis via regulation of NF-κβ mediated inflammation. In addition, BG showed to be effective in the treatment of gastric ulcers, where it has been reported that after aged garlic extract (200 mg/kg) administration in male Wistar rats for 10 consecutive days, oxidative stress, as well as gastric levels of prostaglandin E2, GSH, and NO, were markedly decreased [124]. However, the study by Badr et al. [125] reported a possible mechanism of gastroprotective effects of the aged garlic extract against gastric damage induced by indomethacin in male albino rats, due to BG’s anti-inflammatory actions and antioxidant properties, which released malondialdehyde levels and myeloperoxidase activity and increased total glutathione, SOD, and CAT activities. Similarly, aged garlic extract reduced the intestinal damage of male Wister rats, which was induced by the anti-tumor drug methotrexate. The aged garlic extract suppressed the increase in malondialdehyde in tissue and plasma lactate levels. Thus, aged garlic extract protected against intestinal injury by maintaining cellular integrity [126].

### 6.4. Effects of Black Garlic on Cardiovascular System Diseases

#### 6.4.1. Black Garlic and Platelet Aggregation

The activation of blood platelets plays a vital role in many important physiological and pathological processes, including various arterial phenomena, such as myocardial infarction and strokes [127]. The platelets bind to the exposed collagen, laminin, and von Willebrand factor in the injured vessel wall, referred to as platelet activation. Adenosine-5-diphosphate (ADP) and thrombin can also process the activation. The activated platelets change their shape, remove pseudopodia, release granules, and adhere to other platelets, thus commencing the platelet aggregation process. Platelet-activating factor, a cytokine released by neutrophils, monocytes, and platelets, also promotes aggregate formation [128]. Several studies have shown that BG has great potential in inhibiting platelet aggregation (Table 6). Pre-treatment of male Wister rats with aged garlic extracts significantly inhibited the platelet’s ability to aggregate substantially and this effect was noticed on the fourteenth day. Additionally, aged garlic treatment resulted in platelets that responded to collagen by considerably raising both extracellular adenosine triphosphate (ATP) and intra- and extra-cellular thromboxane B2 levels. Furthermore, aged garlic extract treatment inhibited the phosphorylation of collagen-induced ERK, JNK, and p38 in a dose-dependent manner [129]. Fermented garlic is also effective in the inhibition of hypercholesterolemia and platelet aggregation. It was observed that the oral administration of fermented garlic (300 mg/kg) in male Sprague-Dawley rats once a day, along with a hypercholesterolemia diet for one month, inhibited collagen and ADP-induced platelet aggregation and ATP release. Besides, fermented garlic treatment downregulated the expression of sterol regulatory element-binding protein, acetyltransferase-2, and 3-hydroxy-3-methylglutaryl coenzyme A [130]. Very similar results were also reported by Irfan et al. [131]. Several clinical trials have been conducted in the recent years to investigate the cardioprotective effects of BG. Most of the studies have been found to have positive effects. In a study, Seo et al. [132] showed that a 12 week aged garlic extract regimen with regular exercise reduced cardiovascular risk factors in postmenopausal by decreasing the body weight, body mass index, total cholesterol, LDL-C, malondialdehyde, and homocysteine levels. In another study, Jung et al. [133] investigated whether aged black garlic supplementation could improve blood lipid profile in mild hypercholesterolemia patients. They found that aged black garlic supplement increased the high-density lipoprotein cholesterol levels with the ratio of low-density lipoprotein cholesterol/apolipoprotein B with a decrease in the levels of apolipoprotein B. While two different clinical trials showed that aged garlic extract exerted selective inhibition on platelet aggregation, adhesion, and platelet functions. Aged garlic also significantly suppressed the total percentage and initial platelet aggregation rate [134,135]. Another study, where patients with metabolic syndrome received 1.2g/day of aged garlic extract for 12 weeks, showed that plasma adiponectin levels were increased after the administration of aged garlic extract [136]. On the other hand, a very recent clinical trial study conducted by Wlosinska et al. [137], who reported that daily supplementation of aged garlic extract (2400 mg) for one year lowered the IL-6 in female patients with a low-risk profile of cardiovascular diseases.

#### 6.4.2. Black Garlic and Arterial Hypertension

A systolic blood pressure (SBP) of 140 mm Hg or over and a diastolic blood pressure (DBP) of 90 mm Hg or higher, or both, is considered arterial hypertension. Several risk factors are associated with the development of arterial hypertension. Recent studies have demonstrated that arterial stiffness precedes hypertension as well as causing a gradual increase in SBP [138]. In most cases, suffering from a debilitating disease such as arterial hypertension results in premature death. On the other hand, the renin-angiotensin system (RAS) is an associated hormone group that works together to regulate blood pressure, cardiovascular, and kidney function. It is well documented that RAS dysregulation might be linked to arterial hypertension, cardiovascular, and kidney disease [139]. According to the classical concept of the RAS pathway for arterial hypertension, renin cleaves its substrate, angiotensinogen, to create the inactive peptide angiotensin I, which is then converted to angiotensin II by endothelial angiotensin-converting enzyme (ACE). The most extensive activation of angiotensin II by ACE occurs in the lungs. Angiotensin II acts as a vasoconstrictor and stimulates the production of aldosterone from the adrenal gland, resulting in sodium retention and increased blood pressure [139]. Numerous studies showed the potential effects of BG on decreasing arterial hypertension and inhibiting ACE (Table 6).

Ried et al. [140] carried out a clinical trial on 88 patients with uncontrolled arterial hypertension patients. After 12 weeks of study, the findings indicated that aged garlic extract significantly reduced mean blood pressure along with arterial stiffness, mean arterial pressure, central blood pressure, central pulse pressure, pulse wave velocity, and augmentation pressure. Aged garlic extract administration for 12 weeks in 49 participants with uncontrolled arterial hypertension was also found effective in reducing blood pressure and had the potential to improve inflammation, arterial stiffness, and enhanced gut microbial profile [141]. In another clinical trial, a systolic blood pressure reduction was observed in 50 patients with uncontrolled pressure, the average decline being 10.2 ± 4.3 mm Hg [142].

Similarly, animal studies have shown the potential effects of BG extracts and their bioactive compounds on the inhibition of endothelial ACE activity (Table 6). In 2010, Castro et al. [143] conducted a study on spontaneously hypertensive rats to observe the effects of allyl methyl sulfide (AMS) and DAS on the growth and migration of cultured aortic smooth muscle cells. The authors found that both AMS and DAS inhibited aortic smooth muscle cell angiotensin II-stimulated cell-cycle progression and migration. In addition, the inhibitory actions of these compounds are possibly connected with the suppression of extracellular signal-regulated kinase 1/2 phosphorylation and prevention of the cell cycle inhibitor p27 downregulation. In a separate study, Yu et al. [144] reported that BG extract was the most active in ACE inhibition with the lowest IC_50_ value (0.04 mg/mL) compared to raw garlic extract. Having said that, the authors also identified two Amadori compounds in BG extract, specifically N-(1-deoxy-D-fructos-1-yl)-l-arginine and N-(1-deoxy-D-fructos-1-yl)-l-methionine, which were probably attributed to ACE inhibitory activity. It has also been reported by Jang et al. [145], in their study, that ACE inhibitory effects of the BG extract were greater (88.8%) than normal garlic extract (52.7%). Additionally, a recent study indicated that high levels of oxygen free radicals (OFRs) in the hypothalamic paraventricular nucleus (PVN) contribute to the potentiation of the vasoconstrictor angiotensin II and the cardiac sympathetic afferent reflex (CSAR), which may result in hypertension [146]. Based on their hypothesis, Miao et al. [147] explored the antihypertensive effect of BG in spontaneously hypertensive rats (SHRs) and demonstrated that BG exerted a potential antihypertensive effect through OFRs in the plasma and PVN of SHRs. The authors concluded that bioactive compounds of BG may be transported across the blood–brain barrier of SHRs into PVN to scavenge excess OFRs, hence lowering blood pressure by inhibiting angiotensin II and CSAR potentiation.

However, there is still not enough data to suggest BG for the treatment of hypertensive patients as a standard clinical therapy. More properly designed and analyzed trials are needed for a definite conclusion.

#### 6.4.3. Black Garlic and Atherosclerosis

Atherosclerosis occurs as a result of fat, cholesterol, and other substances interacting within the cellular components of the arterial wall. These deposits are called plaques. Over time, these plaques can eventually narrow or totally block the arteries, causing complications throughout the body [148]. Numerous pre-clinical and clinical investigations have demonstrated and confirmed BG’s efficacy in preventing and treating atherosclerosis (Table 6). Early studies on experimental ApoE-KO mice showed that aged garlic extract suppressed the development of atherosclerosis in ApoE-KO mice by 22% in a 12-week study. Treatment with aged garlic extract significantly suppressed the increase in serum concentrations of total cholesterol, triglycerides (TGs), and reduced the relative abundance of CD11b^+^ cells in ApoE-KO mice [149]. In another study, aged garlic extract exerts anti-atherogenic effects in cholesterol-fed rabbits through reduction of fatty steak development, vessel wall cholesterol accumulation, and the development of fibro-fatty plaques as well as lowering the progression of coronary artery calcification (CAC) [150]. In a randomized double-blind placebo-controlled trial, aged garlic extract significantly inhibited CAC progression, lowered the levels of IL-6 glucose, and blood pressure in 104 patients at increased risk of cardiovascular events after taking 2400 mg/daily of aged garlic extract for one year [151]. Another randomized clinical trial showed that a combination of aged garlic extract and co-enzyme Q10 suppressed the CAC progression and significantly decreased carbon reactive protein levels [152]. In 2009, a placebo-controlled, double-blind, randomized study involved 65 intermediate-risk patients and supplied them a capsule containing aged garlic extract with vitamin B6, vitamin B12, folic acid, and L-arginine for 12 months. At the end of the clinical trial, the data suggested that aged garlic extract with supplements decreased TGs, LDL-C, homocysteine, immunoglobulin G, immunoglobulin M autoantibodies to malondialdehyde-low density lipoprotein and apolipoprotein B-immune complexes while high-density lipoprotein, oxidized phospholipids/apolipoprotein B and lipoprotein were significantly increased [153]. Furthermore, a study of 60 asymptomatic patients showed that aged garlic extract with supplements decreased the levels of epicardial adipose tissue, pericardial adipose tissue, periaortic adipose tissue, and subcutaneous adipose tissue after 12 months of treatment. Thus, aged garlic extract with supplements reduced the metabolic risk and the severity of coronary artery calcification by suppressing the progression rate of adipose tissue volume [154].

### 6.5. Effects of Black Garlic on Neurodegenerative Diseases

There has been an alarming increase in the incidence of neurodegenerative disorders. Although treatments for Alzheimer’s, Parkinson’s, and Huntington’s diseases have shown significant advancement, the pathophysiology of these diseases is not fully clear yet. It is expected that an early decrease in hallmarks of degenerative processes (e.g., apoptosis, inflammation, oxidative stress, and immune dysfunction) could delay the onset and reduce the symptoms of neurodegenerative diseases by providing human subjects with neuroprotective agents. A number of studies have been documented aged black garlic and SAC as potential agents to protect the brain against neurodegeneration (Table 7). In one preliminary study, pretreatment of the Alzheimer’s Swedish double mutant mouse model (Tg2576) with aged garlic extract (2%) reduced cerebral plaques, detergent soluble, and detergent-resistant Aβ-species with concomitantly increased soluble amyloid precursor protein α, reduced inflammation, and conformational change in tau. Thus, the authors suggested that changes in tau phosphorylation appear to be mediated by glycogen synthase kinase 3 beta [156]. In another study, Jeong and his colleagues found that aged garlic extract ameliorated against Aβ-induced neurotoxicity in PC12 mice by showing 2,2′-azino-bis (3-ethylbenzthiazoline-6-sulfonic acid) diammonium salt radical scavenging activity, malondialdehyde inhibitory effect, and reducing intracellular ROS accumulation. In addition, aged garlic extract also improved cognitive impairment in ICR mice by attenuating Aβ-induced learning and memory deficits [157]. BG may also prevent cognitive decline by protecting neurons from Aβ-induced neurotoxicity, and apoptosis, thereby avoiding neuronal death caused by ischemia or reperfusion and improving memory retention and learning ability. In this context, Wichai et al. [158] conducted a dose-dependent study of aged garlic extract on Aβ-induced neurotoxicity rats for 8 weeks. The authors found that aged garlic extract increased the activities of SOD, GPx, and reduced the malondialdehyde levels. Thus, these findings suggested that aged garlic extract improved cognitive dysfunction through its antioxidant effects. Interestingly, another study proposed a possible mechanism, where aged garlic extract attenuated the impairment of working memory via the modification of cholinergic neurons, vesicular glutamate transporter 1, and glutamate decarboxylase in the hippocampus of Aβ-induced rats [159]. Several studies have mentioned that oxidative stress is generated by an increase in the buildup of ROS, which has been linked to the promotion of aging and the pathogenesis of several neurodegenerative disorders. Under all such conditions, additional antioxidant protection is required. Ray et al. [160] showed that aged garlic extract and SAC were able to prevent oxidative insults to neuron cells (~ 80%) from ROS-mediated damage, and preserved the levels of presynaptic protein (~70%), such as SNAP25, in the neuronal culture (PC12) and amyloid precursor protein-transgenic (APP-Tg) mouse model. Furthermore, in a lipopolysaccharide-activated murine BV-2 microglial cell study, aged garlic extract and its compound *N*-α-(1-deoxy-D-fructos-1-yl)-L-arginine attenuated neuroinflammatory responses by suppressing the NO production and regulating the expression of multiple protein targets associated with oxidative stress [161].

On the other hand, Parkinson’s disease (PD) is the second most prevalent age-related neurodegenerative disease with an unclear etiology. Oxidative stress and inflammation play a critical role in the onset and progression of Parkinson’s disease. BG and its compounds have been shown to protect cells from apoptosis, mitochondrial dysfunction, inflammation, and oxidative stress. Very limited studies have shown the effects of BG on Parkinson’s disease. One study evaluated the effects of SAC against oxidative stress in 1-methyl-4-phenylpyridinium ion (MPP^+^)-induced parkinsonism in C57BL/6J mice. SAC (125 mg/kg i.p.) administrated to the mice for 17 days significantly ameliorated MPP^+^-induced lipid peroxidation, ROS production, loss of dopamine in the striatum, and improved locomotion deficits [162]. However, there is a need for more animal studies and clinical trials to confirm the therapeutic effects of BG on Parkinson’s diseases.

Cerebral ischemia, including stroke, is the world’s leading cause of death and disability. It causes neural function loss and irreparable damage to the brain. Cerebral ischemia is defined by a shortage of oxygen, a halt of blood flow, and an influx of calcium, resulting in an increase in ROS, mitochondrial malfunction, and neuronal death. A study conducted by Ashafaq et al. [163] demonstrated the ability of SAC to mitigate oxidative damage and improve neurological deficit in a rat model of focal cerebral ischemia. SAC treatment significantly reduced ischemic lesion volume, suppressed neuronal loss, inhibited glial fibrillary acidic protein, and inducible nitric oxide expression. In another study, both aged garlic extract and SAC showed the ability to induce neuroprotection by controlling ROS. This effect was related to their antioxidant ability in increasing the mRNA expression levels of glucose transporter 3 and glutamate-cysteine ligase catalytic subunit in rats with transient focal cerebral ischemia. Similarly, Cervantes et al. [164] reported that treatment of a rat model of stroke with aged garlic extracts protected against ischemia-induced brain damage. The effects of aged garlic extract attributed to the decrease in the mRNA expression of N-methyl-D-aspartate receptor subunits after ischemia prevents ischemia-induced reduction in mitochondrial potential and ATP synthesis. SAC, at a dose of 300 mg/kg, treated cerebral ischemia in male Wister rats. The results suggested that SAC diminished cerebral ischemia-induced mitochondrial dysfunctions in the hippocampus by restoring the GSH, glucose 6-phosphate dehydrogenase, and decreasing lipid peroxidation, protein carbonyl, hydrogen peroxide (H_2_O_2_) content as well as brain edema [165]. The neuroprotective effect of aged garlic extract is not only due to its anti-oxidant effect but also to its anti-inflammatory effect. In a study, Colin-Gonzalez et al. [166] showed that aged garlic extract administration attenuated the increase in the levels of oxidative stress markers such as 8-hydroxy-2-deoxyguanosine and TNF-α with COX-2 protein.

### 6.6. Effects of Black Garlic on Cancer Diseases

Cancer is one of the leading causes of death throughout the world. The treatment of cancer varies and usually includes surgery, radiation therapy, and chemotherapy. The utilization of natural products in cancer therapy is a rapidly growing field of study. Many pre-clinical studies have been conducted to investigate the anti-cancer effects of BG due to its various health benefits and pharmacological effects including anti-oxidant, anti-inflammatory, apoptosis induction, anti-proliferation, and anti-angiogenesis (Table 8). In vitro studies have shown that BG is very capable of suppressing cancer cell growth and proliferation in multiple cancer cell lines. For example, one in vitro study demonstrated the ability of aged black garlic extract to suppress colon cancer cell growth (HT29) and promoted apoptosis by inhibiting the P13KAkt pathway [32]. Moreover, aged black garlic extract was found to be able to inhibit gastric cell growth both in vitro and in vivo animal studies. An in vitro study revealed dose-dependent apoptosis in SGC-7901 cells treated with aged black garlic extract while an in vivo animal study showed the anti-cancer effects of aged black garlic extract such as tumor growth suppression in tumor-bearing mice. The authors suggested that the anti-cancer effects of aged black garlic extracts may be attributed to their antioxidant and immunomodulatory properties [33]. Similar inhibition properties of BG extract and its compound, SAMC, were reported in the case of lung cancer Lewis cells and thyroid cancer cell line (HPACC-8305C), respectively [168,169]. Numerous studies have demonstrated BG’s anti-tumor properties through suppression of cell proliferation against colon cancer and gastric cancer. In 2014, Jikihara and his collaborators treated F344 rats and DLD-1 human colon cancer cells with aged garlic extract. The results showed inhibition of proliferative activity in adenoma and adenocarcinoma lesions. In addition, aged garlic extract delayed cell cycle progression by downregulating cyclin B1 and cyclin-dependent kinase 1 expression via inactivation of NF-κβ but did not induce apoptosis [170]. Through the inhibition of matrix metalloproteinase-2, matrix metalloproteinase-9, and repressed levels of claudin proteins, water extract of aged black garlic may also inhibit tumor metastasis and invasion in gastric cell (AGS) [171]. Aged garlic extract has also been reported to have a chemo-sensitizing effect on doxorubicin (DOX) in the human breast cancer cell (MCF-7). Aged garlic extract improves the cytotoxic effect of DOX on MCF-7 cells, most likely through apoptosis induction, enhanced intracellular DOX accumulation, and suppression of P-glycoprotein activity [172]. Recently, the anticancer effects of SAMC, one of the BG compounds, have been demonstrated on prostate cancer, liver cancer, bladder cancer, and ovarian cancer. It was demonstrated that SAMC could: (i) show positive effects against prostate cancer cells by altering prostate biomarker expression and utilizing testosterone to restore E-cadherin’s expression [173,174,175,176,177,178]; (ii) promote MAPK inhibitor-induced apoptosis by activating the TGF-β signaling pathway [179]; (iii) inhibit the survival, invasion, and migration of bladder cancer cells through the inactivation of inhibitor of differentiation-1 pathway [180]; and (iv) suppress both the proliferation and distant metastasis of epithelial ovarian cancer cells [181].

## 7. Conclusions and Future Perspectives

Although BG has been recognized in Asia since ancient times, there is a growing interest in its intake around the world because of its singular organoleptic qualities and bioactive effects. The questionable origin of BG and the lack of a standard technique of preparation has led to the lack of a quality index, which makes it challenging to obtain a standardized BG product. There are several aspects that can directly affect BG’s sensory and nutritional properties, including processing technologies and processing variables such as time, temperature, humidity, pH, as well as type of pre-treatment. In particular, some processes and reactions are critical to the properties of the final BG, notably the Millard reaction. Moreover, the interactions between garlic compounds as well as microbes during the aging process also affect the properties of BG. As reported in this review, a large number of studies have been conducted to prepare the BG under different processing conditions. As a result, variations in phytochemical contents of BG were also noticed after the Millard reaction. It is clear from this review that more research is needed to find out the optimal conditions for BG processing, the roles of key compounds on biological properties, as well as to establish a clear relation between chemical and sensory characteristics of BG.

Numerous pre-clinical and clinical studies have provided solid evidence to support the therapeutic potential of BG consumption in various preparations in the treatment of various human diseases. The aged garlic extract is the most studied formulation among the available preparation and has shown effective pharmacological activity. The present review suggests that the therapeutic effects of BG are mainly attributed to its antioxidant, immunomodulatory, anti-inflammatory, anticancer, anti-diabetic, anti-obesity, digestive system protective, hepatoprotective, cardiovascular protective, neuroprotective, and nephroprotective activities. BG’s therapeutic benefits appear to be mediated by the regulation of several signaling molecules. However, in many cases, the underlying mechanisms are unknown because of the complexity of the disorders. Indeed, there are only very few and inconsistent outcomes from human studies, presumably because of variances in BG preparations, unknown active substances, and their bioavailability, as well as small sample size. Therefore, the hypothesized in vitro and in vivo animal studies should be further verified in human studies to provide a deeper understanding of BG’s therapeutic potential.

## Figures and Tables

**Figure 1 molecules-26-05028-f001:**
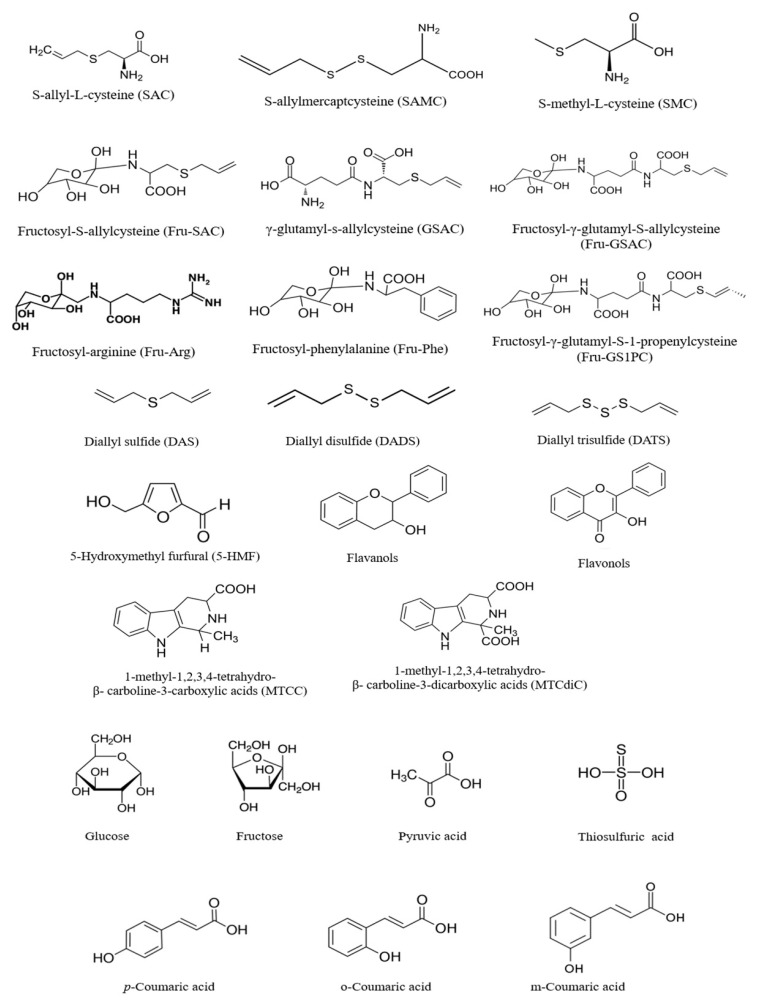
Chemical structures of compounds predominately found in black garlic.

**Table 1 molecules-26-05028-t001:** Processing conditions of black garlic.

Temperature	Relative Humidity	Durations (Days)	Results	Ref.
60 °C	80%	69	Temp.	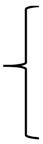	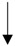	Moisture	[41]
70 °C	33	Allicin
80 °C	24		
90 °C	12	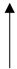	HMF
Total phenols
Total acids
70 °C	90%	35		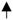 Redness 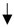 Brightness and yellowness	[46]
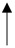	Crude fat, crude protein, total sugar,
total pyruvate, glucose, amino acids
70 °C	90%	21		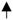 Redness 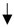 Lightness and yellowness 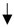 pH	[40]
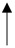	Total polyphenol, total flavonoid,
total acidity, reducing sugar
65 °C	70%	16	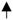	Polyphenol content (85 °C, 70% RH)	[42]
75 °C	75%		
78 °C	80%	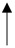	Reducing sugar, total sugar
85%	total acids, 5-HMF (75 °C, 85% RH)
72 °C	~90%	33	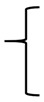	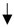	pH	[44]
	Temp.
75 °C	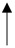	Polyphenols, antioxidant,
*Browning intensity*
78 °C		
90 °C (1st step)		2	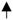 Moisture (4th step)	[43]
80 °C (2nd step)	4	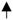 Total phenolic and flavonoids (4th step)
60 °C (3rd step)	4	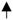 Total pyruvate and thiosulfate (4th step)
40 °C (4th step)	1	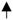 Mineral content (4th step)
65–80 °C	70–80%	30–40	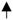 SAC	[45]
70 °C	90%	10	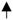 SAC (15 days)	[47]
15
20
40 °C	70%	45	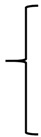	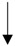	Moisture	[16]
pH
	Temp.
55 °C	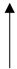	*Browning density*
70 °C	SAC
85 °C	*Antioxidant activity*

Temp.: temperature; 5-HMF: 5-Hydroxymethylfurfural; SAC: s-allyl cysteine.

**Table 2 molecules-26-05028-t002:** Phytochemical contents in fresh garlic and black garlic.

Phytochemical	Total Content in Fresh Garlic	Total Content in Black Garlic	Ref.
Amino acid	843.11 ± 3.75 mg/100 g	167.65 ± 1.08−363.10 ± 1.05 mg/100 g	[73]
	121.38 ± 4.72 mg/100 g	60.84 ± 5.75−108.66 ± 12.95 mg/100 g	[46]
	19.43 ± 0.01 mg/g FM	14.86 ± 0.01 mg/g FM	[79]
	57.66 mg/g sample powder	44.01 mg/g sample powder	[80]
	1943.77 ± 161.22 mg/100 g	1486.65 ± 112.62 mg/100 g	[81]
	1528.75 ± 0.83 mg/100 g	1931.13 ± 175.48 mg/100 g	[40]
Minerals	1173.50 ± 2.43 mg/100 g	1314.68 ± 2.76 mg/100 g	[73]
	15,908 mg/kg dry powder	13,227.41 mg/kg dry powder	[80]
	567.88 ± 4.48 mg/100 g	969.12 ± 19.31 mg/100 g	[40]
Reducing sugar	58.37 ± 1.54 mg/100 g	394.52 ± 3.29 mg/100 g	[40]
	1.52 ± 0.01 g/kg	12.42 ± 0.85−16.07 ± 0.38 g/kg	[46]
	295.54 ± 2.01 mg/100 g	754.51 ± 4.05−4726.04 ± 15.74 mg/100 g	[73]
	5.9 ± 0.8 g/kg DM	472.4 ± 46.5 g/kg DM	[18]
Total phenol	1000 µg/g	8200 µg/g	[63]
	18 mg GAE/kg DM	80–140 mg GAE/kg DM	[82]
	89468.55 mg QE/kg dry basis	101,328.71–157,312.77 mg QE/kg dry basis	[83]
	14 mg GAE/g	20–60 mg GAE/g	[46]
	15,200 mg GAE/kg	24,050 mg GAE/kg	[65]
	5150 mg GAE/kg DM	14,900 mg GAE/kg DM	[44]
Total flavonoid	2348.65 mg GAE/kg dry basis	3825.51−27,191.38 mg GAE/kg dry basis	[83]
	3.22 ± 0.07 mg RE/g	5.38 ± 0.06−15.70 ± 2.11 mg RE/g	[46]
	30.03 mg/kg DM	30–105 mg/kg DM	[82]
	0.25 mg/100 g	0.70 mg/100 g	[43]
	0.20 mg/100 g FG water extract	0.50 mg/100 g BG water extract	[84]
	1.40 µg CAE/mg	1.92 µg CAE/mg	[85]
SAC	42.7 µg/g DM	656.5 µg/g DM	[86]
	2.5 mg/g DW	8.05 mg/g DW	[87]
		95.07 ± 1.84−427.05 ± 3.56 µg/g	[88]
		85.46 ± 0.81−124.67 ± 1.61 µg/g	[16]
	73.5 ± 12.5 µg/g DW	242.3 ± 6.1 µg/g DW	[89]
	2.4 mg/100 g	19.4 mg/100 g	[90]
5-HMF		0.25 ± 0.04 g/kg FM	[65]
		4.82 ± 0.06 g/kg	[41]
		6–8 g/kg	[42]
Melanoidin	˂ 0.2 OD FM	~2 OD FM	[75]
Ash	73.59 ± 0.89 mg/100 g	75.36 ± 0.02−114.36 ± 8.65 mg/100 g	[73]
	0.92 ± 0.62 %	1.81 ± 0.05 %	[40]
Volatile compounds	49.76 µg/g	39.04−100.46 µg/g	[83]
Organic acid	16.70 ± 0.61 g/kg DM	64.50 ± 7.55 g/kg DM	[41]
Alkaloid	Trace amount	30-fold increase of FM	[91]
Lipid	0.20 ± 0.01%	0.60 ± 0.11%	[40]
	0.1%	0.30%	[90]
Carbohydrate	30%	50%	[90]
Protein	0.9%	1.2 ± 0.1%	[40]
	8.4%	9.5%	[90]
Vitamin	6632.91 ± 18.62 mg/kg	7618.24 ± 28.47–9,010.44 ± 30.61 mg/kg	[70]
Pyruvate	19.01 ± 0.3 mmol/100 g	28.05 ± 0.3 mmol/100 g	[40]
	49.05 ± 1.2 mmol/100 g	246.02 ± 2.4 mmol/100 g	[20]
Thiosufate	6.50 ± 0.29 µM/g	91.22 ± 0.54 µM/g	[20]

FG: fresh garlic; BG: black garlic; DM: dry matter; DW: dry weight; FM: fresh matter; GAE: gallic acid equivalent; QE: quercetin equivalent; RE: rutin equivalent; CAE: catechin equivalents; OD: optical density for an absorbance at 420 nm.

**Table 3 molecules-26-05028-t003:** Effects of black garlic on metabolic disorders.

Diseases	Products	Subjects/Cell Line/Animal Model	Outcomes	Mode of Action	Ref.
**Diabetes mellitus**	Black garlic juice (BGJ)	Male C57BL/6J mice	SAC-enriched BGJ counteracted STZ239-induced diabetes and β-cell failure in mice.	Improved glutathione antioxidant system, increased leptin and adiponectin secretion. Inhibited hepatic gluconeogenesis and NF-κβ-mediated inflammatory signaling.	[89]
	Aged garlic	Male Sprague-Dawley rats	Ameliorated oxidative stress and other complications of diabetes.	Decreased body weight, blood glucose, serum cholesterol, triglycerides, and fructosamine.	[97]
	Aged black garlic (ABG)	C57BL/KsJ-db/db mice	ABG improved insulin sensitivity and dyslipidemia in db/db mice.	Decreased serum glucose, total cholesterol, triglyceride and increased HDL-C levels.	[30]
	Black garlic powder (BGP)	Male Wister rats	BGP lowered blood glucose, prevented glycogen in the liver, and improved lipid metabolism.	Lowered glycosylated Hb, and total cholesterol and increased HDL-C. BGP increased the activity of GOT, GPT, γ-GTP in serum.	[98]
	Aged black garlic (ABG)	C57BL/KsL-db/db mice	ABG prevented diabetic complications through antioxidant activity.	Decreased TBARS levels, elevated the activities of SOD, GSH-Px, and CAT.	[99]
	Aged garlic (AG) and S-allyl cysteine (SAC)	BSA or lysozyme	AG + SAC prevented the formation of advanced glycation end products.		[107]
	Aged garlic (AG)	Sprague-Dawley rats	AG exhibited ameliorative action on indicators of diabetes.	Decreased blood glucose, GHb, and lipid peroxidation. Markedly increased serum insulin, serum triglyceride elevation, and total cholesterol.	[96]
**Obesity**	Aged garlic (AG)	51 healthy adults with obesity Study period: 6 weeks	AG prevented the development of chronic diseases associated with low-grade inflammation.	Decreased TNF-α, IL-6, blood LDL levels.	[104]
	Fermented garlic by lactic acid bacteria (FBLA)	Male C57BL/6J mice	FBLA ameliorated diet-induced obesity by inhibiting adipose tissue hypertrophy by suppressing adipogenesis.	Reduced body weight, TG, TC, retroperitoneal, epididymal, and mesenteric adipose tissue mass. Downregulated mRNA protein expression of PPARγ, C/EBPα, and lipogenic proteins, including SREBP-1c, FAS, and SCD-1.	[102]
	Aged garlic	Male Sprague-Dawley rats	Modified the adipose weight and improved the oxidative stress.	Decreased Body weight gain, visceral, epididymal fat, and TBARS levels.	[101]
	Black garlic	Male Wister rats	Ameliorated diet-induced obesity via regulating adipogenesis, adipokine biosynthesis, and lipolysis.	Upregulated AMPK, FOXO1, Sirt1, ATGL, HSL, perilipin, ACO, CPT-1, UCP1, adiponectin, and PPAR α. Downregulated CD36, SREBP-1c, ACC, FAS, and SCD1.	[105]
	Aged garlic	Sprague Dawley rats	Exhibited anti-obesity, cholesterol-lowering, and anti-inflammatory effects.	Reduced body weight, visceral fat, liver weight, total cholesterol, low-density lipoprotein, and C-reactive protein.	[103]
	Aged black garlic	Male Sprague-Dawley rats	Improved the body weight gain and dyslipidemia through the suppression of body fat and alteration in lipid profiles and antioxidant defense system.	Decreased the body weight, adipose tissue weight, TC, TG, and increased oxidized GSH and LPO in the serum.	[29]
	Black garlic (BG)	Male Wister rats	BG ameliorated obesity induced by a HFD in rats.	Decreased body weight, tissue weight of liver, epididymal fat, peritoneal fat, serum triglycerides, hepatic lipid profile, GSSG, and enhanced TEAC, GSH, GRd, and GPx.	[25]
	Aged black garlic	3T3-L_1_ preadipocytes	Exhibited anti-lipogenic and lipolytic effects.	Reduced protein expression of PPARγ, HSL, and Ser-pHSL levels.	[106]

SAC: s-allyl cysteine; NF-κβ: nuclear factor-κβ; HDL-C: high-density lipoprotein cholesterol; GOT: glutamic oxaloacetic transaminase; GPT: glutamic pyruvic transaminase; γ-GTP: γ- glutamyl transpeptidase; SOD: superoxide dismutase; GSH-Px: glutathione peroxidase; CAT: catalase; GHb: glycated hemoglobin; STZ: streptozotocin; TBARS: thiobarbituric acid reactive substances; TNF-α: tumor necrosis factor-α; IL-6: interleukin-6; LDL: low density lipoprotein; TG: triglyceride; TC: total cholesterol; mRNA: messenger ribonucleic acid; C/EBPα: CCAAT/enhancer-binding protein α; UCP1: uncoupling protein 1; CPT-1: carnitine palmitoyl transferase 1; ACO: acyl-coenzyme A oxidase; HSL: hormone sensitive lipase; ATGL: adipose triglyceride lipase; Sirt1: sirtuin 1; FOXO1: forkhead box O1; PPARα: peroxisome proliferator-activated receptor α; AMPK: AMP-activated protein kinase; SREBP-1c: sterol regulatory element binding protein-1c; ACC: acetyl-CoA carboxylase; FAS: fatty acid synthase; SCD1: stearoyl-CoA desaturase-1; HFD: high fat diet; LPO: lipid peroxidation; GSSG: glutathione disulfide; TEAC: trolox equivalent antioxidant capacity; GSH: glutathione; GRd: glutathione reductase; GPx: glutathione peroxidase; PPARγ: proliferator activated receptor γ; Ser-pHSL: serum-phosphorylated HSL.

**Table 4 molecules-26-05028-t004:** Effects of black garlic on genitourinary tract diseases.

Diseases	Products	Subjects/Cell Line/Animal Model	Outcomes	Mode of Action	Ref.
Nephrotoxicity	Aged garlic (AG)	Male Wister rats	AG prevented gentamicin-induced nephrotoxicity.	Decreased the oxidative stress and preserved the activities of Mn-SOD, GPx, and GR	[108]
Diabetic nephropathy (DNP)	Aged garlic (AG)	Albino Wistar rats	AG significantly decreased albumin levels in urine, blood urea nitrogen contents, and increased urine urea nitrogen contents.	The protective effect of AG on DNP due to its anti-glycation, hypolipidemic effects.	[109]
Kidney damage	Aged garlic (AG)	Albino Wistar rats	AG rescued ethephon-induced kidney damage.	Activation of Nrf2 and inhibition of inflammation and apoptotic response.	[110]
Kidney injury	Aged black garlic (ABG)	Male Sprague-Dawley rats	ABG ameliorated colistin-induced acute kidney injury in rats.	Reduced the levels of oxidative stress biomarkers such as 8-hydroxydeoxyguanosine and malondialdehyde. Lowered the levels of NF-κβ, inducible NO synthase, COX-2, and TGF-β1, and also restored SOD, CAT, and GSH levels.	[111]

Mn-SOD: manganese superoxide dismutase; GPx: glutathione peroxidase; GR: glutathione reductase; Nrf2: nuclear factor-erythroid factor 2-related factor 2; NF-κβ: nuclear factor-κβ; NO: nitric oxide; COX-2: cyclooxygenase-2; TGF-β1: transforming growth factor beta 1; SOD: superoxide dismutase; CAT: catalase; GSH: glutathione.

**Table 5 molecules-26-05028-t005:** Effects of black garlic on digestive disorders.

Diseases	Products	Subjects/Cell Line/Animal Model	Outcomes	Mode of Action	Ref.
**Liver**	Aged garlic	C57BL/6 mice	Modulation of glycometabolism, lipometabolism, oxidative stress, and inflammation.	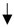 ALT, AST, TC, LDL-C, MDA	[114]
Black garlic (BG)	Wister rats	BG protected against oxidative damage caused by CCl_4_ -induced liver injury.	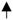	SOD, GSH-Px	[112]
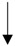	ALT, AST, LDH,
and ALP
Lactic acid-fermented garlic	Wister rats	Protected against oxidative liver injury.	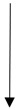	ALP, AST, and ALT	[115]
ATP depletion
TNF-α, IL-1β
Apoptosis
(BCL-2, Bax, Cascape-3)
Fermented blackgarlic	C57BL/6J mice	Improved the effects on fatty liver.	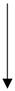	AST, ALT	[26]
Total cholesterol,
LDL/V-LDL-cholesterol,
triglyceride contents
Black garlic	ICR mice	Inhibited CCl_4_-induced hepatic injury by inhibiting lipid peroxidation and inflammation.	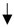 ALT, AST, ALP, and MDA 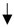 TNF-α and IL-1β 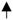 SOD, GSH-Px, GSH-Rd	[28]
Aged black garlic	Sprague-Dawley rats	Exhibited protective effects against chronic alcohol-induced liver damage.	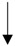	AST, ALT, ALP,	[113]
and LDH

	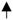 GSH-Px, GR, and CAT
Aged black garlic	Sprague-Dawley rats	Showed hepatoprotective effects against liver injury.	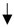 AST, and ALTNo effects on ALP.	[27]
**Inflammatory**	Aged black garlic	RAW 264.7 cells	Suppressed the expression of classical mitogen-activated protein kinases (MAPKs) (ERK1/2 and p38 MAPK) in LPS-stimulated macrophage cells.	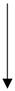	NO,	[22]
prostaglandin E2
IL-6, TNF-α, IL-1β
iNOS, COX-2
Aged black garlic	U-937 cells	Inhibited expression of COX-2 and production of prostaglandin E2.	Inactivation of NF-κβ.	[116]
Aged black garlic	HES cells	Reduced cell proliferation and attenuated the expression of ICAM-1 and VCAM-1.	Inhibition of the ERK, JNK signaling pathways, ROS formation, NF-κβ, and AP-1 transcription factors.	[119]
Aged black garlic	RAW 264.7 macrophages	Exerted anti-inflammatory effects.	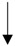	NO, TNF-α,	[120]
prostaglandin E2
Fermented black garlic	RAW 264.7 cells	Exhibited anti-inflammatory effects.	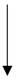	NO, TNF-α,	[21]
prostaglandin E2
IL-1β, IL-6
5-HNF	RAW 264.7 cells	Exerted anti-inflammatory effects.	Inhibition of MAPK, NF-κβ and Akt/mTOR pathways.	[118]
5-HNF	HUVE cells	Prevented TNF-α-induced monocytic cell adhesion to HUVE cells.	Suppressed vascular cell adhesion molecule-1 expression, reactive oxygen species generation and NF-κβ activation.	[117]
**Gastro-intestinal motility**	Black garlic	Sprague–Dawley rats	Effectively promoted gastrointestinal motility and defection.	Stimulated gastrointestinal peristalsis, enhanced gastrointestinal tract emptying, and promoted defecation.	[121]
**Laxative** **effects**	Mixed black garlicbeverage	BALB/c mice	Exhibited an obvious laxative effect, which improved the intestinal flora of mice.	Showed a relatively high ink-propelling rate, increased defection time, *E. coli* and *Enterococci.*	[122]
**Gastroesopha-geal reflux disease**	Black garlic	Sprague-Dawley rats	Showed protective effect on reflux esophagitis.	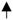 SOD, CAT 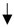 TNF-α, IL-6	[123]
**Gastric ulcer**	Aged garlic	Male Wistar rats	Prevented the indomethacin-induced ulcer.	Reduced oxidative stress.Increased gastric levels of PGE2, GSH, and NO.	[124]
**Gastric damage**	Aged garlic	Malealbino rats	Heal the gastric mucosal injury induced by indomethacin.	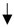 MDA, MPO 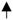 tGSH, SOD, CAT	[125]
**Intestinal damage**	Aged garlic	Male Wistar albino rats	Reduced Intestinal damaged induced by anti-tumor drug methotrexate in the small intestine.	Aged garlic halted the MDA increase in tissue and plasma lactate elevations. Thus, protected intestinal damage by preserving cellular integrity.	[126]

AST: aspartate transaminase; ALT: alanine transaminase; TC: total cholesterol; LDL-C: low-density lipoprotein-cholesterol; SOD: superoxide dismutase; GSH-Px: glutathione peroxidase; LDH: lactate dehydrogenase; ALP: alkaline phosphatase; ATP: adenosine triphosphate; TNF-α: tumor necrosis factor; IL-1β: interleukin-1β; BCL-2: B-cell lymphoma 2; Bax: BCL-2-associated X protein; MDA: malondialdehyde; CAT: catalase; GR: glutathione reductase; MAPK: mitogen-activated protein kinase; LDL/V-LDL: low density lipoprotein/very-low-density lipoprotein; GSH-Rd: glutathione reductase; IL-6: interleukin-6; ERK: extracellular-signal-regulated kinase; JNK: c-Jun N-terminal kinase; NO: nitric oxide; LPS: lipopolysaccharides; VCAM-1: vascular cell adhesion protein-1; ICAM-1: intercellular adhesion molecule-1; COX-2: cyclooxygenase-2; iNOS: inducible nitric oxide synthase; AP-1: activator protein-1; NF-κβ: nuclear factor-κβ; PGE2: prostaglandin E2; GSH: glutathione; MPO:**** myeloperoxidase; tGSH: total glutathione.

**Table 6 molecules-26-05028-t006:** Effects of black garlic on cardiovascular system diseases.

Diseases	Products	Subjects/Cell Line/Animal Model	Outcomes	Mode of Action	Ref.
**Platelet** **Aggregation**	Aged garlic	30 participantsStudy period:12 weeks	Reduced cardiovascular risk factors	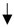 BW, BMI, TC, LDL-C, MDA, 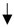 Homocysteine 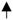 TG	[132]
Agedblack garlic	28 participantsStudy period:12 weeks	Reduced atherogenic markers.	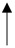		[133]
HDL-CRatio of LDL-C/apo-B

Aged garlic (AG)	34 participantsStudy period:44 weeks	AG exerted selective inhibition on platelet aggregation and adhesion, platelet functions.		[134]
Aged garlic (AG)	23 participantsStudy period:13 weeks	AG substantially suppressed the total percentage and initial platelet aggregation rate.		[135]
Aged garlic (AG)	43 participants Study period: 24 weeks	AG increased plasma adiponectin levels.		[136]
Aged garlic(AG)	31 participants Study period:12 months	AG lowered IL-6 in females with a low-risk profile of the cardiovascular disease.		[137]
Aged garlic (AG)	Male Wistar rats	Suppressed the platelet aggregation by changing the functional property of the platelets.	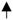 Extracellular ATP 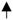 Extra- and intracellular TXB_2_Suppressed the phosphorylation of collagen-induced ERK, p38, and JNK.	[129]
Fermented garlic	Male Sprague-Dawley rats	Ameliorated hypercholesterolemia and inhibited platelet activation.	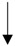		[131]
TG, SERBP-2,ACAT-2, HMG-CoA

Fermented garlic (FG)	Male Sprague-Dawley rats	FG significantly inhibited platelet aggregation and granule secretion in hypercholesterolemic rats.	Inhibited collagen and ADP-induced platelet aggregation and ATP release. Downregulated the expression of SERBP, ACAT-2, and HMG-CoA.	[130]
**Arterial Hypertension**	Aged garlic (AG)	88 patients with uncontrolled arterial hypertension.Study period:12 weeks	Reduced mean blood pressure along with arterial stiffness, mean arterial pressure, central blood pressure, central pulse pressure, pulse-wave velocity, and augmentation pressure.		[140]
Aged garlic (AG)	49 participants with uncontrolled arterial hypertension.Study period:12 weeks	AG was effective in reducing blood pressure and had the potential to improve inflammation, arterial stiffness, and gut microbial profile.		[141]
Aged garlic	9 patients Study period:12 weeks	Mean systolic blood pressure was significantly reduced.		[155]
Aged garlic	50 patients Study period:12 weeks	Systolic blood pressure wasreduced on an average of 10.2 ± 4.3 mm Hg.		[142]
**Hypertension** **related to RAS**	Allyl methylSulfide (AMS)anddiallyl sulfide (DAS)	Male spontaneously hypertensive rats (SHRs)	AMS and DAS inhibited aortic smooth muscle cell angiotensin II-stimulated cell-cycle progression and migration.	The outcome was probably mediated via upregulation of the growth suppressor p27 and the attenuation of ERK 1/2 phosphorilation.	[143]
Black garlic(BG)	Male spontaneously hypertensive rats (SHRs)	BG exerted a potential antihypertensive effect through OFRs in the plasma and PVN of SHRs.	Declined high blood pressure via abolishing the potentiation of angiotensin II and CSAR.	[147]
Black garlic (BG)	Rabbit lung ACE	BG was the most active in ACE inhibition with the lowest IC_50_ value (0.04 mg/mL).	Amadori compounds (Fru-Arg and Fru-Met) were probably attributed to ACE inhibitory activity.	[144]
Black garlic		ACE inhibitory effects of the black garlic extract were greater (88.8%) than normal garlic extract (52.7%)		[145]
**Atherosclerosis**	Aged garlic (AG)	104 patients Study period:12 months	AG suppressed the atherosclerosis progression.	Inhibited CAC progression, lowered the levels of IL-6, glucose, and blood pressure.	[151]
Aged garlic (AG)and coenzyme Q10 (CoQ10)	65 patientsStudy period:12 months	AG + CoQ10 reduced the progression of coronary atherosclerosis.	Lowered CAC progression and CRP levels.	[152]
Aged garlic	60 patientsStudy period:12 months	Reduced the progression rate of adipose tissue volumes related to CAC.	Decreased the levels of EAT, PAT, PaAT, and SAT.	[154]
Aged garlic supplemented with B vitamins, folic acid, and L-arginine	65 patientsStudy period:12 months	Improved oxidative biomarkers, vascular function, and reduced progression of atherosclerosis.	Lowered CAC progression. Decreased TG, LDL-C, homocysteine, IgG and IgM autoantibodies to MDA-LDL and apoB-immune complexes. Increased HDL, OxPL/apoB, and LP.	[153]
Aged garlic (AG)	ApoE-KO mice	AG suppressed the development of atherosclerosis.	Suppressed the increase in serum concentrations of TC, TG and reduced the relative abundance of CD11b^+^ cells.	[149]
Aged garlic(AG)	New Zealand white rabbit	AG protected the onset of atherosclerosis.	Reduced fatty streak development, vessel wall cholesterol accumulation, and the development of fibro-fatty plaques.	[150]

BW: body weight; BMI: body mass index; TC: total cholesterol; LDL-C: low-density lipoprotein cholesterol; MDA: malondialdehyde; TG: triglyceride; HDL-C: high-density lipoprotein cholesterol; LDL-C/apo-B: low-density lipoprotein cholesterol/apolipoprotein B; ATP: adenosine triphosphate; TXB_2_: thromboxane B2; ERK: extracellular-signal-regulated kinase; JNK: c-Jun N-terminal kinase; SERBP-2: sterol regulatory element binding protein-2; ACAT-2: acetyltransferase-2; HMG-CoA: 3-hydroxy-3-methylglutaryl coenzyme A; ADP: adenosine diphosphate; IL-6:interleukin-6; RAS: renin-angiotensin system; OFRs: oxygen free radicals; PVN: paraventricular nucleus; CSAR: cardiac sympathetic afferent reflex; ACE: angiotensin-converting enzyme; Fru-Arg: N-(1-deoxy-D-fructos-1-yl)-l-arginine; Fru-Met: N-(1-deoxy-D-fructos-1-yl)-l-methionine; CAC: coronary artery calcification; CRP: c-reactive protein; EAT: epicardial adipose tissue; PAT: pericardial adipose tissue; PaAT: periaortic adipose tissue; SAT: subcutaneous adipose tissue; IgG: immunoglobulin G; IgM: immunoglobulin M; MDA-LDL: malondialdehyde-low-density lipoprotein; HDL: high-density lipoprotein; OxPL/apoB: oxidized phospholipids/apolipoprotein B; LP: lipoprotein.

**Table 7 molecules-26-05028-t007:** Effects of black garlic on neurodegeneration diseases.

Diseases	Products	Subjects/Cell Line/Animal Model	Outcomes	Mode of Action	Ref.
**Alzheimer’s**	Aged garlic (AG)	Transgenic model Tg2576	AG reduced cerebral plaques, detergent soluble and detergent resistant (fibrillar) Aβ-species with concomitantly increased α-cleaved sAPPα, reduced inflammation and conformational change in tau.	The observed change in tau phosphorylation appears to involve GSK-3β.	[156]
Agedgarlic (AG)	PC12 cellICR mice	AG ameliorated against Aβ-induced neurotoxicity and cognitive impairment.	AG showed ABTS radical scavenging activity, MDA inhibitory effect and reduced intracellular ROS accumulation.	[157]
Aged garlic (AG)	Male Wistar rat	AG ameliorated the cognitive dysfunction in Aβ-induced neurotoxicity rats.	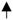 SOD, GPx 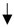 MDA levels	[158]
Aged garlic (AG) and Fru-Arg	Murine BV-2 microglial cell	AG and Fru-Arg attenuated neuroinflammatory responses.	Suppressed the production of NO. Regulated the expression of multiple protein targets associated with oxidative stress.	[161]
Aged garlic(AG)	Male Wistar rats	AG significantly improved the working memory and tended to improve the reference memory in cognitively impaired rats.	Modified the cholinergic neurons, VGLUT1, and GAD in the hippocampus of Aβ-induced rats.	[159]
Aged garlic(AG)and S-allyl-L-cysteine (SAC)	PC12 cells,Tg2576transgenic mice	Both AG and SAC treatment protected neuronal cells from ROS-mediated oxidative insults and preserved the levels of pre-synaptic protein such as SNAP25.		[160]
**Parkinson’s**	S-allyl-L-cysteine (SAC)	C57BL/6J mice	SAC protected against oxidative stress in 1-methyl-4 phenylpyridinium-induced parkinsonism in mice.	Ameliorated MPP^+^-induced lipid peroxidation, ROS production, loss of dopamine in striatum, and improved locomotion deficits.	[162]
**Cerebral ischemia**	S-allyl-L-cysteine (SAC)	Male Wistar rats	SAC mitigated oxidative damage and improved neurologic deficit.	Reduced ischemic lesion volume, suppressed neuronal loss, and inhibited glial fibrillary acidic protein and inducible nitric oxide expression.	[163]
Aged garlic(AG)and S-allyl-L-cysteine (SAC)	Male Wistar rats	Both AG and SAC treatment induced neuroprotection.	Increased GLUT3 and GCLC mRNA expression levels.	[167]
S-allyl-L-cysteine (SAC)	Male Wistar rats	SAC diminished cerebral ischemia-induced mitochondrial dysfunctions in hippocampus.	Restored GSH and G6-PD. Decreased LPO, PC, and H_2_O_2_ content as well as the brain edema.	[165]
Aged garlic (AG)	Male Wistar rats	AG attenuated the cerebral ischemia-induced inflammation.	Attenuated the increase in the levels of 8-OHdG, TNF-α, and COX-2 protein.	[166]
Aged garlic(AG)	Male Wistar rats	AG protected against ischemia-induced brain damage.	Decreased mRNA expression of NMDA receptor subunits after ischemia. Prevented ischemia-induced reduction in mitochondrial potential and in ATP synthesis.	[164]

Aβ: β-amyloid; sAPPα: soluble amyloid precursor protein α; GSK-3β: glycogen synthase kinase 3 beta; ABTS: 2,2 -azino-bis (3-ethylbenzothiazoline-6-sulfonic acid; MDA: malondialdehyde; ROS: reactive oxygen species; SOD: superoxide dismutase; GPx: glutathione peroxidase; Fru-Arg: N-α-(1-deoxy-D-fructos-1-yl)-L-arginine; NO: nitric oxide; VGLUT1:vesicular glutamate transporter 1; GAD: glutamate decarboxylase; SNAP25: synaptosomal associated protein of 25 kDa; MPP^+^: 1-methyl-4-phenylpyridinium ion; GLUT3: glucose transporter 3; GCLC: glutamate cysteine ligase catalytic subunit; GSH: glutathione; G6-PD: glucose 6-phosphate dehydrogenase; LPO: lipid peroxidation; PC: protein carbonyl; 8-OHdG: 8-hydroxy-2-deoxyguanosine; TNF-α: tumor necrosis factor-α; COX-2: cyclooxygenase-2; mRNA: messenger ribonucleic acid; NMDA: N-methyl-D-aspartate; ATP: adenosine triphosphate.

**Table 8 molecules-26-05028-t008:** Effects of black garlic on different cancers.

Diseases	Products	Subjects/Cell Line/Animal Model	Outcomes	Mode of Action	Ref.
**Colon** **cancer**	Aged black garlic (ABG)	HT29 cell	ABG inhibited colon cancer cell growth	ABG reduced HT29 cell growth and promoted apoptosis by inhibiting the PI3KAkt pathway.	[32]
	Aged garlic (AG)	DLD-1 cell and F344 rats	AG inhibited 1,2-dimethylhydrazine-induced colon tumor development.	AG delayed cell cycle progression by downregulating cyclin B1 and cdk1 expression via inactivation of NF-κB but did not induce apoptosis.	[170]
**Prostate** **cancer**	SAMC	LNCaP cell, PC-3, DU 145 cells	SAMC showed positive effects against prostate cancer cells.	Rescued GSH deficits, altered prostate biomarker expression and utilized testosterone, restored the expression of E-cadherin.	[173,174,175,176,177,178]
**Gastric** **cancer**	Aged black garlic (ABG)	AGS cells	ABG treatment inhibited tumor metastasis and invasion.	ABG increased the tightness of tight junction. Inhibited the activities of MMP-2 and -9 in AGS cells. Repressed the levels of claudin proteins.	[171]
	Aged black garlic (ABG)	SGC-7901 cells	ABG induced inhibition of gastric cancer cell growth.	Increased the superoxide dismutases, glutathione peroxidase, interleukin-2, and indices of spleen and thymus in Kunming mice.	[33]
**Breast** **cancer**	Aged garlic (AG)	MCF-7	Exhibited a chemosensitizing effect.	Induction of apoptosis, enhanced intracellular DOX accumulation, inhibition of P-gp activity.	[58]
**Liver** **cancer**	SAMC	HepG2 cells	SAMC promoted MAPK inhibitor-induced apoptosis by activating the TGF-β signaling pathway.	Activated TGF-β1, TβRII, psmad2/3, smad4 and smad7 signaling.	[179]
**Bladder** **cancer**	SAMC	MGH-U1 cells	SAMC inhibited the survival, invasion, and migration of bladder cancer cells.	Inactivated Id-1 pathway.	[180]
**Thyroid** **cancer**	SAMC	HPACC-8305C	SAMC inhibited the growth of HPACC-8305C cells.	Induction of apoptotic cell death. Inhibited telomerase activity.	[168]
**Lung** **cancer**	Black garlic	Lewis cells	Inhibited the growth of lung cancer cells.	Affected the expression of Bax and BCL-2	[169]
**Ovarian** **cancer**	SAMC	HO8910, HO8910PM, and SKOV3	SAMC suppressed both the proliferation and distant metastasis of epithelial ovarian cancers cells.	Down-regulated the survivin gene in HO8910PM cells with small interference RNA (siRNA). Decreased invasiveness of tumor cells.	[181]

cdk1: cyclin-dependent kinase 1; NF-κβ: nuclear factor-κβ; GSH: glutathione; MMP-2: matrix metalloproteinase-2; MMP-9: matrix metalloproteinase-9; DOX- doxorubicin; P-gp: P-glycoprotein; MAPK: mitogen-activated protein kinase; TGF-β1: transforming growth factor beta 1; TβRII: type II TGF-β receptor; p-samd 2/3: phosphorylated- suppressor of mothers against decapentaplegic 2/3; smad 4: suppressor of mothers against decapentaplegic homolog 4; smad7: suppressor of mothers against decapentaplegic homolog 7; Id-1: inhibitor of differentiation-1; BCL-2: B-cell lymphoma 2; Bax: BCL-2-associated X protein.

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
