# Peer review of "Black Garlic and Its Bioactive Compounds on Human Health Diseases: A Review"

_molecules, 2021, doi:10.3390/molecules26165028_

Round 1

Reviewer 1 Report

This review literature aimed at summarizing the clinical and preclinical studies on the potential beneficial properties of black garlic on some chronic diseases. Although the idea of this review was original, the paper presents several biomedical inaccuracies. Furthermore, the paper is not easy to read as it is a list of the studies published on the topic without there being a personal comment by the authors.

Major comments:

The definition of diabetes mellitus is incorrect, it should be revised and expanded.

The section on chronic kidney disease presents a number of conceptual errors; the authors generically insert the term "kidney disorders" in the text, without specifying what they refer to, ie glomerular pathologies, tubulo-interstitial pathologies, vascular pathologies, infectious pathologies.

In addition, there are several errors in renal function biomarkers; the authors incorrectly indicate in the text the term "deterioration of serum creatinine levels", they should have correctly written "deterioration of renal function monitored by standardized biomarkers such as ..."

Minor Comments

In the line 41, correct “markers” with “outcomes”

In the line 473, the authors insert the term "diabetes hyperglycemia" which is incorrect.

In the line 478, correct the term "indicators of diabetes" with "glyco-metabolic biomarkers".

Correct in all text “ hypertension”  with “ arterial hypertension”

Author Response

Comment#1: The definition of diabetes mellitus is incorrect, it should be revised and expanded.

Response#1: Thank you for the comment, to prevent mistake, this part is rewritten according to the reviewer's suggestion with a reference in line no. 455-457.

Diabetes mellitus is a heterogeneous form of metabolic disorder characterized by chronic hyperglycemia with impaired carbohydrate, fat, and protein metabolism resulting from defects in insulin secretion, insulin action, or both.

Comment#2: The section on chronic kidney disease presents a number of conceptual errors; the authors generically insert the term "kidney disorders" in the text, without specifying what they refer to, i.e., glomerular pathologies, tubulo-interstitial pathologies, vascular pathologies, infectious pathologies.

Response#2:  Thank you for the good suggestion. We revised this section and added some more clear information in line no. 541, 542.  

Only a few research have examined the role of aged garlic extract in the treatment of kidney disorders, more precisely those caused by glomerular, tubulo-interstitial, or infectious pathologies (Table 5).                

Comment#3:  In addition, there are several errors in renal function biomarkers; the authors incorrectly indicate in the text the term "deterioration of serum creatinine levels", they should have correctly written "deterioration of renal function monitored by standardized biomarkers such as ...".

Response#3:   Thank you for the valuable suggestion. The correction has been done accordingly in line no. 556, 557.

Treatment with aged black garlic extract prevented deterioration of renal function monitored by standardized biomarkers such as the serum levels of creatinine and blood urea nitrogen.

Comment#4:     In the line 41, correct “markers” with “outcomes”

Response#4:    Thank you for good comment. The word was revised again in line no. 47 according to your suggestion.

Notably, plant-based dietary pattern foods have improved different intermediary outcomes with a lower incidence of chronic diseases, including cancer, cardiovascular, neurodegenerative diseases, modulate intestinal microbiota composition and functionality with a positive human health impact.

Comment#5:  In the line 473, the authors insert the term "diabetes hyperglycemia" which is incorrect.

Response#5:   Sorry for the mistake and thank you for suggestion. The term has been modified in line no 458.

Clinical and experimental studies have demonstrated that chronic hyperglycemia is a major source of oxidative stress and that elevated free radicals play a key role in diabetes mellitus pathogenesis and complications.

Comment#6: In the line 478, correct the term "indicators of diabetes" with "glyco-metabolic biomarkers".

Response#6:   Thank you for the suggestion. The correction has been done in lines 462, 463.

For example, aged black garlic extract exhibited ameliorative action on glyco-metabolic biomarkers in streptozotocin-induced diabetic rats by significantly decreasing blood glucose, glycated hemoglobin (GHb), and markedly increased serum insulin.

Comment#7:  Correct in all text “hypertension” with “arterial hypertension”

Response#7:  The suggestion has been taken into consideration and modified this section by combining the other reviewer comment in line 727-731, 742, 743, 747, Table 10.

Several risk factors are associated with the development of hypertension. In this context, arterial stiffness is one of the principal causes of increasing SBP with advancing years and in patients with arterial hypertension [142]. In most situations, having a debilitating dis-ease such as arterial hypertension results in premature death.

Ried et al. [144] carried out a clinical trial on 88 patients with uncontrolled arterial hypertension patients.

Aged garlic extract administration for 12 weeks in 49 participants with uncontrolled arterial hypertension patients was also found effective in reducing blood pressure and had the potential to improve inflammation, arterial stiffness, and enhanced gut microbial profile.

Arterial Hypertension

Aged garlic (AG)

88 patients with uncontrolled

arterial hypertension.

Study period:

12 weeks

Reduced mean blood pressure along with arterial stiffness, mean arterial pressure, central blood pressure, central pulse pressure, pulse-wave velocity, and augmentation pressure.

[144]

Aged garlic (AG)

49 participants with uncontrolled

arterial hypertension.

Study period:

12 weeks

AG was effective in reducing blood pressure and had the potential to improve inflammation, arterial stiffness, and gut microbial profile.

[145]

Reviewer 2 Report

Reviewer’s comments (molecules-1302682):

The review article by Ahmed and Wang focuses on black garlic and its bioactive molecules on human diseases. The authors summarized (and discussed) the pre-clinical and clinical studies on black garlic and related components. Finally, the mode of action and the various bioactivities of black garlic components are reviewed regarding their therapeutic potential in the treatment of specific human diseases.

In my opinion, this is an interesting, and rather complete review article on black garlic and related compounds. The manuscript is overall well-written and discussed, but could be significantly improved for clarity. Indeed, some parts are confusing and might be revised, highlighting the most important properties of black garlic and related molecules.

I have a couple of additional concerns (to potentially improve the quality of the manuscript), as follows:

  1. The title of manuscript should be modified (as there are many bioactive compounds in black garlic, not just one);
  2. According to the reported effects of black garlic (antioxidant/reductant, antimicrobial, immunomodulatory, anti-inflammatory, anti-cancer, anti-diabetic, anti-obesity, as well as cardiovascular, hepato, neuro, renal and intestinal protections), one would suggest a direct action on the renin-angiotensin system (RAS). The authors might discuss on this point;
  3. There are 12 tables in this review article, which is too high. It would have been appropriate to better condense/summarize the most relevant data in a limited number of tables;
  4. The manuscript should be revised by a native English speaker for grammar and typos.

Author Response

Comment#1: The title of manuscript should be modified (as there are many bioactive compounds in black garlic, not just one);

Response#1:   Thank you for the suggestion. We modified the title accordingly;

                        Black garlic and its bioactive compounds on human health 
                        diseases: A review

Comment#2:  According to the reported effects of black garlic (antioxidant/reductant, antimicrobial, immunomodulatory, anti-inflammatory, anti-cancer, anti-diabetic, anti-obesity, as well as cardiovascular, hepato, neuro, renal and intestinal protections), one would suggest a direct action on the renin-angiotensin system (RAS). The authors might discuss on this point;

Response#2:   We appreciate your constructive criticisms. We discussed this point separately in the hypertension section and highlighted it in the line 731 to 741, 752 to 775, and Table 10.

On the other hand, the renin-angiotensin system (RAS) is an associated hormone group that works together to regulate blood pressure, cardiovascular, and kidney function. It is well documented that RAS dysregulation might be linked to hypertension, cardiovascular, and kidney disease [143]. According to the classical concept of the RAS pathway for hypertension, renin cleaves its substrate, angiotensinogen (AGT), to create the inactive peptide angiotensin I, which is then converted to angiotensin II by endothelial angiotensin-converting enzyme (ACE). The most extensive activation of angiotensin II by ACE occurs in the lung. Angiotensin II acts as a vasoconstrictor and stimulates the production of aldosterone from the adrenal gland, resulting in sodium retention and increased blood pressure [143]. Numerous studies showed the potential effects of black garlic on decreasing arterial hypertension and inhibiting ACE (Table 10).

Similarly, animal studies have shown the potential effects of black garlic extracts (BGE) and its bioactive compounds on the inhibition of endothelial angiotensin-converting enzyme activity. In 2010, Castro et al. [147] conducted a study on spontaneously hypertensive rats to observe the effects of allyl methyl sulfide (AMS) and diallyl sulfide (DAS) on the growth and migration of cultured aortic smooth muscle cells. They have found that both AMS and DAS inhibited aortic smooth muscle cell angiotensin II-stimulated cell-cycle progression and migration. Rather than that, the inhibitory actions of these compounds are possibly connected with the suppression of extracellular signal-regulated kinase 1/2 phosphorylation and prevention of the cell cycle inhibitor p27 downregulation. In a separate study, Yu et al. [148] reported that BGE was the most active in ACE inhibition with the lowest IC50 value (0.04 mg/mL) compared to raw garlic extract. Having said that, the authors also identified two amadori compounds in BGE, specifically Fru-Arg and Fru-Met, which were probably attributed to ACE inhibitory activity. It has also been reported by Jang et al. [149] that ACE inhibitory effects of the BGE were greater (88.8%) than normal garlic extract (52.7%) in their study. Additionally, recent study indicates that high levels of oxygen free radicals (OFRs) in the hypothalamic paraventricular nucleus (PVN) contribute to the potentiation of the vasoconstrictor angiotensin II and the cardiac sympathetic afferent reflex (CSAR), which may result in hypertension[150]. Based on the hypothesis, Miao et al. [151] explored the antihypertensive effect of black garlic in spontaneously hypertensive rats (SHRs) and demonstrated that BG exerted a potential antihypertensive effect through OFRs in the plasma and PVN of SHRs. They concluded that bioactive compounds of BG may be transported across the blood-brain barrier of SHRs into PVN to scavenge excess OFRs, hence lowering blood pressure by inhibiting angiotensin II and CSAR potentiation.

Hypertension

related to RAS

Allyl methyl

Sulfide (AMS)

and

diallyl sulfide (DAS)

Male spontaneously hypertensive rats (SHRs)

AMS and DAS inhibited aortic smooth muscle cell angiotensin II-stimulated cell-cycle progression and migration.

The outcome was probably mediated via upregulation of the growth suppressor p27 and the attenuation of ERK 1/2 phosphorilation.

[147]

Black garlic

(BG)

Male spontaneously hypertensive rats (SHRs)

BG exerted a potential antihypertensive effect through OFRs in the plasma and PVN of SHRs.

Declined high blood pressure via abolishing the potentiation of angiotensin II and CSAR.

[151]

Black garlic (BG)

Rabbit lung ACE

BG was the most active in ACE inhibition with the lowest IC50 value (0.04 mg/mL).

Amadori compounds (Fru-Arg and Fru-Met) were probably attributed to ACE

inhibitory activity.  

[148]

Black garlic

ACE inhibitory effects of the black garlic extract were greater (88.8%) than normal garlic extract (52.7%)

[149]

RAS: renin-angiotensin system; OFRs: Oxygen free radicals; PVN: Paraventricular nucleus; CSAR: Cardiac sympathetic afferent reflex; ACE: angiotensin-converting enzyme; Fru-Arg: N-(1-deoxy-D-fructos-1-yl)-l-arginine; Fru-Met: N-(1-deoxy-D-fructos-1-yl)-l-methionine;

Comment#3:  There are 12 tables in this review article, which is too high. It would have been appropriate to better condense/summarize the most relevant data in a limited number of tables;

Response#3: Thank you for your suggestion. Yes, we will summarize the most relevant data in 8 tables. Especially combined the disorders into 6 tables.

Comment#4: The manuscript should be revised by a native English speaker for grammar and typos.

Response#4: Thank you for suggestion. The whole manuscript was cross-checked for grammatical errors by native English speakers, and all the changes were highlighted in yellow.

Round 2

Reviewer 1 Report

The review, although improved compared to the previous version, continues to present some critical issues especially in the biomedical section. In this section, there are still several inaccuracies.

Despite having requested in the previous review to include some comments on the articles cited, in addition to listing them, no changes have been made in this regard.

Major comments:

  • Rephrase lines 34-36 (page 2) which are difficult to understand and have errors.
  • Lines 56-58 and 76-79 (page 2) express the same concept, remove this repetition.
  • Correct paragraph 6 title as follows: "Impact of black garlic on ....".
  • Rephase lines 556-558. This sentence has some conceptual errors.
  • Correct paragraph 6.3 title as follows: " on digestive diseases".
  • Line 613, insert some examples of inflammatory diseases to which the authors refer
  • Please insert the abbreviations list
  • Correct throughout the text the acronyms that must be entered the first time they are cited, then use the acronym for the following times.
  • Correct paragraph 6.4.2 title as follows: " …and arterial hypertension".
  • In the text correct “ hypertension” with “ arterial hypertension”
  • Lines 795-797, revised the definition of atherosclerosis
  • Correct paragraph 6.5 title as follows: " …neurodegenerative diseases".
  • The authors write in vitro, in vivo and clinical studies: in vivo studies include clinical studies. They should specify in vivo animal or human studies; the latter include clinical studies. Correct this concept throughout the text

Minor Comments:

  • In the abstract, correct “renal protection “ with “ nephroprotection”
  • In the abstract, line 23, delete “disease”.
  • Line 42, page 2, correct “age-related disorders” with “ age-related pathologies”
  • Line 218, remove the capital letter from the word hydroxymethylfurfural.
  • Line 464: correct “fat” with “lipid”.
  • Line 474: correct “per oxidation” with “peroxidation”.
  • Line 505: correct “adipose tissue weight” with “adipose tissue mass”.
  • Title 6.3.2., remove the capital letter from the word inflammatory.
  • In the text, in vivo and in vivo must always be written in italics
  • Lines 694-696 rephrase the sentence. It presents some errors.
  • Line 695, correct disorders with “phenomena”
  • Line 730, remove the capital letter from the word adiponectin.
  • Line 741, delete and in patients with….
  • Lines 741-743 rephrase the sentence.
  • Line 760, delete “patients”
  • Line 789, correct “ hypertension patients” with “ hypertensive patients”
  • Line 851, remove the capital letter from the word immune.
  • Lines 971-973, correct this sentence
  • Revised lines 1004-1018, present some errors

Author Response

Thank you for your constructive comments. We have answered each of your points below.

Revisions that are made:

Major comments:

Comment#1

Rephrase lines 34-36 (page 2) which are difficult to understand and have errors.

Response#1

Thank you for the suggestion. Lines 34-36 have been rephrased to make it more understandable.

With the prevalence of chronic diseases and their associated pathological complications, health has become the top of scientific research priorities, with the goal of finding novel foods and tactics to tackle such public health burden.

Comment#2

Lines 56-58 and 76-79 (page 2) express the same concept, remove this repetition.

Response#2

We apologize for the repetition error. We have checked these lines carefully and removed line 56-58 from the manuscript.

For instance, multiple studies showed the bioactivities of BG, including antioxidation[20], anti-inflammation [21–23], anti-obesity[24,25], hepatoprotection[26–28], hypolipidemia[29,30], anti-cancer[31–33], anti-allergy[34,35], immunomodulation[15,36], cardiovascular prevention[37,38], and neurodegenerative protection[39].

Comment#3

Correct paragraph 6 title as follows: "Impact of black garlic on ....".

Response#3

Thank you so much for your suggestion. Correction has been made in paragraph 6 title.

Impact of black garlic on health promotion and diseases treatment

Comment#4

Rephrase lines 556-558. This sentence has some conceptual errors.

Response#4

Thank you for your constructive comment. We have rephrased these lines again in lines 541-543 to avoid the conceptual errors.

In the case of diabetic nephropathy disease, aged garlic extract significantly decreased albumin levels in urine, blood urea nitrogen contents, and increased urine urea nitrogen contents in diabetic rats.

Comment#5

Correct paragraph 6.3 title as follows: " on digestive diseases"

Response#5

Thank you for the suggestion. The title has been modified based on your suggestion.

Effects of black garlic on digestive diseases

Comment#6

Line 613, insert some examples of inflammatory diseases to which the authors refer

Response#6

Thank you so much for the valuable suggestion. Examples of inflammatory diseases have been inserted in lines 595-596.

Numerous in vitro investigations have demonstrated that BG has significant potential for treating a variety of diseases related to inflammation such as lethal sepsis, endometriosis, rheumatoid arthritis, inflammatory bowel diseases etc.

Comment#7

Please insert the abbreviations list.

Response#7

We appreciate your suggestion. An abbreviation list has inserted after the conclusions and future perspectives section in lines 996-997.

Comment#8

Correct throughout the text the acronyms that must be entered the first time they are cited, then use the acronym for the following times.

Response#8

Thank you for your constructive suggestion. We have revised the whole manuscript again and made corrections on the acronyms according to your suggestion following marked by yellow color.

Comment#9

Correct paragraph 6.4.2 title as follows: " …and arterial hypertension".

Response#9

Thank you. We have revised the title again and made correction as below:

Black garlic and arterial hypertension

Comment#10

In the text correct “hypertension” with “arterial hypertension”.

Response#10

Thank you for the suggestion. We made correction “hypertension” word with “arterial hypertension” in lines 716-717, 723-724, 737.

A systolic blood pressure (SBP) of 140 mm Hg or over and a diastolic blood pressure (DBP) of 90 mm Hg or higher, or both, is considered arterial hypertension.

Several risk factors are associated with the development of arterial hypertension.

It is well documented that RAS dysregulation might be linked to arterial hypertension, cardiovascular, and kidney disease.

According to the classical concept of the RAS pathway for arterial hypertension, renin cleaves its substrate, angiotensinogen, to create the inactive peptide angiotensin I, which is then converted to angiotensin II by endothelial angiotensin-converting enzyme (ACE).

Comment#11

Lines 795-797, revised the definition of atherosclerosis.

Response#11

Thank you. We have revised the atherosclerosis definition again with a reference.

Atherosclerosis occurs as a result of fat, cholesterol and other substances interacting within the cellular components of the arterial wall. These deposits called plaques. Over time, these plaques can eventually narrow or totally block the arteries, causing complications throughout the body[148]

Comment#12

Correct paragraph 6.5 title as follows: " …neurodegenerative diseases".

Response#12

Thank you. The tittle has been modified according to your suggestion.

Effects of black garlic on neurodegenerative diseases

Comment#13

The authors write in vitro, in vivo and clinical studies: in vivo studies include clinical studies. They should specify in vivo animal or human studies; the latter include clinical studies. Correct this concept throughout the text.

Response#13

Thank you for your advice. We have carefully checked the entire manuscript and corrections have been marked in yellow color to clear the concept throughout text.

Minor comments:

Comment#1

In the abstract, correct “renal protection “with “nephroprotection”

Response#1

Thank you. Correction has been made in line 19.

Most of these benefits can be attributed to its anti-oxidation, anti-inflammation, anti-obesity, hepatoprotection, hypolipidemia, anti-cancer, anti-allergy, immunomodulation, nephroprotection, cardiovascular protection, and neuroprotection.

Comment#2

In the abstract, line 23, delete “disease”.

Response#2

Thank you for the suggestion.

Thus, in this review, we summarized the pre-clinical and clinical studies of BG and its bioactive compounds on human health along with diverse bioactivity, a related mode of action, and also future challenges.

Comment#3

Line 42, page 2, correct “age-related disorders” with “age-related pathologies”

Response#3

Thank you. We revised the line 42 according to your comment.

An increasing amount of scientific data suggests that dietary interventions, particularly those provides high levels of plant foods, can prevent or delay the growth of chronic age-related pathologies by safeguarding the body's critical physiological systems.

Comment#4

Line 218, remove the capital letter from the word hydroxymethylfurfural.

Response#4

We apologize for the error.

Similarly, hydroxymethylfurfural (HMF) is one of the major antioxidant ingredients in BG.

Comment#5

Line 464: correct “fat” with “lipid”.

Response#5

Thank you. The word “fat” replaced with “lipid” in line 454.

Diabetes mellitus is a heterogeneous form of metabolic disorder characterized by chronic hyperglycemia with impaired carbohydrate, lipid, and protein metabolism resulting from defects in insulin secretion, insulin action, or both [95].

Comment#6

Line 474: correct “per oxidation” with “peroxidation”.

Response#6

We apologize for the typing error.

Additionally, aged garlic extract significantly attenuated the elevation of serum triglyceride, total cholesterol, and lowered lipid peroxidation in liver and kidney tissues[96].

Comment#7

Line 505: correct “adipose tissue weight” with “adipose tissue mass”.

Response#7

Thank you for the suggestion.

BG extracts have been reported for their activity in reducing body weight, adipose tissue mass, serum triglyceride, total cholesterol, low-density lipoprotein, and plasma malondialdehyde in mice with high-fat-diet-induced obesity (Table 3)[25,29,101].

Comment#8

Title 6.3.2., remove the capital letter from the word inflammatory.

Response#8

We apologize for the typing error.

Black garlic and inflammatory diseases

Comment#9

In the text, in vivo and in vivo must always be written in italics.

Response#9

Thank you for your suggestion. We have checked the entire manuscript again and made corrections according to your suggestion following marked by yellow color in lines 611, 916-917, 922.

Comment#10

Lines 694-696 rephrase the sentence. It presents some errors.

Response#10

Thank you for your suggestion. We have revised the lines 694-696 and rephrased the sentence with a reference to avoid the conceptual errors.

The activation of blood platelets plays a vital role in many important physiological and pathological processes, including various arterial phenomena, such as myocardial infarction and strokes [127].

Comment#11

Line 695, correct disorders with “phenomena”

Response#11

Thank you for your suggestion.

The activation of blood platelets plays a vital role in many important physiological and pathological processes, including various arterial phenomena, such as myocardial infarction and strokes [127].

Comment#12

Line 730, remove the capital letter from the word adiponectin.

Response#12

We apologize for the typing error.

Plasma adiponectin levels were increased after administration of aged garlic extract[136].

Comment#13

Line 741, delete and in patients with….

Response#13

Thank you for you suggestion. We have revised the line 741 again and deleted “and in patients with…”.

Recent studies have demonstrated that arterial stiffness precedes hypertension as well as causes gradual increase in SBP.

Comment#14

Lines 741-743 rephrase the sentence.

Response#14

Thank you for the suggestion. Line 741-743 has been rephrased according to your suggestion.

In most cases, suffering from a debilitating disease such as arterial hypertension results in premature death.

Comment#15

Line 760, delete “patients”

Response#15

Thank you. We have revised the line 760 again and deleted the patient word from the line.

Aged garlic extract administration for 12 weeks in 49 participants with uncontrolled arterial hypertension was also found effective in reducing blood pressure and had the potential to improve inflammation, arterial stiffness, and enhanced gut microbial profile[141].

Comment#16

Line 789, correct “hypertension patients” with “hypertensive patients”

Response#16

Thank for your valuable comment.

However, there is still not enough data to suggest BG for the treatment of hypertensive patients as a standard clinical therapy.

Comment#17

Line 851, remove the capital letter from the word immune.

Response#17

We apologize for the typing error.

It is expected that an early decrease in hallmarks of degenerative processes (e.g., apoptosis, inflammation, oxidative stress, and immune dysfunction) could delay the onset and reduce the symptoms of neurodegenerative diseases by providing human subjects with neuroprotective agents.

Comment#18

Lines 971-973, correct this sentence

Response#18

Thank for your advice. We have carefully checked the lines 971-973 again and corrections has been made as follow:

It was demonstrated that SAMC could: i) show positive effects against prostate cancer cells by altering prostate biomarker expression and utilizing testosterone to restore E-cadherin's expression [173–178];

Comment#19

Revised lines 1004-1018, present some errors

Response#19

Thank you so much for your constructive comment. Lines 1004-1018 have been carefully checked again and made corrections to avoid the conceptual errors.

Numerous pre-clinical and clinical studies have provided solid evidence to support the therapeutic potential of BG consumption in various preparations in the treatment of various human diseases. The aged garlic extract is most studied formulation among the available preparation and showed the effective pharmacological activity. The present review suggest that the therapeutic effects of BG mainly attributed to its antioxidant, immunomodulatory, anti-inflammatory, anticancer, anti-diabetic, anti-obesity, digestive system protective, hepatoprotective, cardiovascular protective, neuroprotective, and nephroprotective activities. BG’s therapeutic appears to be mediated by regulation of several signaling molecules. However, in many cases the underlying mechanism is unknown because of complexity of the disorders. Indeed, there are only very few and inconsistent outcomes from human studies, presumably because of variances in BG preparations, unknown active substances and their bioavailability as well as small sample size. Therefore, the hypothesized in vitro, in vivo animal studies should be further verified in human studies to provide a deeper understanding of BG’s therapeutic potential.

Reviewer 2 Report

See 'Comments for Editors'.

Author Response

Thank you for kind review for this manuscript.